# 🐝 BEE: A HIGH-QUALITY CORPUS AND FULL-STACK SUITE TO UNLOCK ADVANCED FULLY OPEN MLLMS

**Yi Zhang**[1,2], **Bolin Ni**[2], **Xin-Sheng Chen**[1], **Heng-Rui Zhang**[1], **Yongming Rao**[2],
**Houwen Peng**[2*], **Qinglin Lu**[2], **Han Hu**[2], **Meng-Hao Guo**[1*] , **Shi-Min Hu**[1†]
[1]Tsinghua University, [2]Tencent Hunyuan Team

🏠: https://open-bee.github.io

## ABSTRACT

Fully open multimodal large language models (MLLMs) currently lag behind proprietary counterparts, primarily due to a significant gap in data quality for supervised fine-tuning (SFT). Existing open-source datasets are often plagued by widespread noise and a critical deficit in complex reasoning data, such as Chain-of-Thought (CoT), which hinders the development of advanced model capabilities. Addressing these challenges, our work makes three primary contributions. First, we introduce Honey-Data-15M, a new SFT dataset comprising approximately 15 million QA pairs, processed through multiple cleaning techniques and enhanced with a novel dual-level (short and long) CoT enrichment strategy. Second, we introduce HoneyPipe, the data curation pipeline, and its underlying framework DataStudio, providing the community with a transparent and adaptable methodology for data curation that moves beyond static dataset releases. Finally, to validate our dataset and pipeline, we train Bee-8B, an 8B model on Honey-Data-15M. Experiments show that Bee-8B establishes a new state-of-the-art (SOTA) for fully open MLLMs, achieving performance that is competitive with, and in some cases surpasses, recent semi-open models such as InternVL3.5-8B. A comprehensive ablation study further dissects the impact of our data curation process, revealing that each stage provides significant performance gains across a wide range of benchmarks. Our work delivers to the community a suite of foundational resources, including: the Honey-Data-15M corpus; the full-stack suite comprising HoneyPipe and DataStudio; training recipes; an evaluation harness; and the model weights. This effort demonstrates that a principled focus on data quality is a key pathway to developing fully open MLLMs that are highly competitive with their semi-open counterparts.

## 1 INTRODUCTION

Massive datasets have been a cornerstone of the success seen in today's powerful multimodal large language models (MLLMs) (Li et al., 2025a; Liu et al., 2023c; Wang et al., 2024b; Hurst et al., 2024; Guo et al., 2025b). However, as the field matures, a new consensus is emerging that data quality is as critical as data quantity, especially for the supervised fine-tuning (SFT) stage (Li et al., 2024d; Wang et al., 2023b). The success of top-tier proprietary models such as Gemini 2.5 (Comanici et al., 2025) and GPT-5 (OpenAI, 2025) relies heavily on highly curated and refined SFT datasets. Such private resources pose a formidable barrier to entry—one that has split the MLLMs field into a distinctly tiered structure: (1) top proprietary models (Guo et al., 2025b; Jaech et al., 2024; Comanici et al., 2025), (2) semi-open MLLMs that release weights but keep data private (Yue et al., 2025b; Bai et al., 2025), and (3) the fully open MLLMs (Li et al., 2025a; Guo et al., 2025c; Deitke et al., 2025), which lag behind the former two. For the fully open community, competing on sheer data volume is not an appropriate strategy. Therefore, focusing on data quality is the most viable path forward.

---

*Project lead. Contact: henryllpeng@tencent.com, gmh@tsinghua.edu.cn
†Corresponding author. Contact: shimin@tsinghua.edu.cn

However, existing open-source SFT datasets suffer from fundamental quality issues that hinder model development. The first of these is widespread data noise (Chen et al., 2024d; Guo et al., 2025c). Existing datasets are frequently contaminated with not only content-level issues such as factual inaccuracies and image-instruction mismatches, but also structural and format-level flaws, such as excessive text repetition, incorrect tags in instructions, and low-quality images with improper sizes or aspect ratios. During training, this noisy data misleads the model into learning spurious correlations, which systematically undermines its core capabilities. This manifests not only as factual hallucinations (Bai et al., 2024) but also as degraded reasoning, poor instruction-following ability, and a tendency to generate low-quality responses. Another critical issue is the gap in complex problem-solving abilities. The capacity of a model for complex reasoning has become a key determinant of its overall capability and a primary manifestation of its advanced intelligence (Kahneman, 2011; Guo et al., 2025e;d; Jaech et al., 2024; Guo et al., 2025a; Yue et al., 2025b). This is clearly demonstrated by leading proprietary and semi-open models, which excel at handling complex instructions, often leveraging long Chain-of-Thought (CoT) processes. The fully open community, however, lags significantly in this area. This weakness is rooted in data deficiencies: the community not only lacks large-scale, high-quality long CoT datasets, but it also remains difficult to identify which instructions actually require these deep, multistep reasoning paths (Kojima et al., 2022). This data deficit represents the primary bottleneck preventing fully open MLLMs from developing the advanced problem-solving skills necessary to compete.

The root of these data quality issues lies not only in the raw data but also in the lack of transparent and reproducible data curation pipelines. Proprietary models benefit from sophisticated, undisclosed recipes for data filtering and cleaning (Guo et al., 2025b; Yue et al., 2025b). A similar opacity exists within the open-source community. Previous work has focused on sharing the final dataset rather than the methodologies behind its creation. Pioneering projects (Li et al., 2025a; Guo et al., 2025c), despite their contributions, often release static datasets. Their curation pipelines—the code, prompts, and filtering logic remain opaque black boxes. This one-off release model is critically flawed. Proprietary efforts constantly refine their internal data recipes. To truly compete, the open-source community needs access to evolving methodologies. The absence of shared, adaptable data curation methods is a fundamental roadblock.

To address these multifaceted challenges, we introduce **Honey-Data-15M**, a large-scale, high-quality SFT dataset designed to serve as a new cornerstone resource for the fully open MLLM community. Its construction is guided by two core principles. First, we conducted a comprehensive data refinement process, collecting from diverse projects and systematically cleaning the corpus to purge widespread data noise, thereby significantly enhancing overall data quality and reliability. Second, we implemented a nuanced, dual-level CoT enrichment strategy that tailors response depth to instruction complexity. For instructions requiring moderate reasoning, we constructed short CoT responses, creating a massive corpus of 12.1M instruction-response pairs. For the most complex instructions, we generated detailed long CoT responses, yielding a high-quality set of 2.9M pairs. This targeted, dual-level approach provides tailored reasoning depth across the entire dataset and inherently solves the critical challenge of identifying which instructions warrant more elaborate, multi-step solutions. Honey-Data-15M was created using our data pipeline, **HoneyPipe**, which is an instantiation of our self-developed data curation framework, **DataStudio**. This pipeline leverages MLLMs to automate the entire curation workflow, from cleaning to enrichment. As a scalable and economical alternative to costly human annotation, this model-driven process makes high-quality data construction feasible for the open-source community.

To validate the effectiveness of our curated Honey-Data-15M, we also contribute a new model to the fully open MLLM ecosystem. Experiments with our final model, **Bee-8B**, trained on the full Honey-Data-15M dataset, establish a new state-of-the-art (SOTA) among **fully open MLLMs**. Its performance is highly competitive, standing on par with, and in some cases surpassing, several recent semi-open models such as InternVL3.5-8B (Wang et al., 2025b). This significant leap is directly attributable to our data curation strategy. A comprehensive ablation study quantifies this impact, showing that our curation process yields significant improvements across multiple benchmarks compared to using the original, unprocessed data. These results confirm that our focus on data quality is a crucial strategy for closing the performance gap between fully open MLLMs and recent semi-open models.

In summary, our key contributions are threefold:

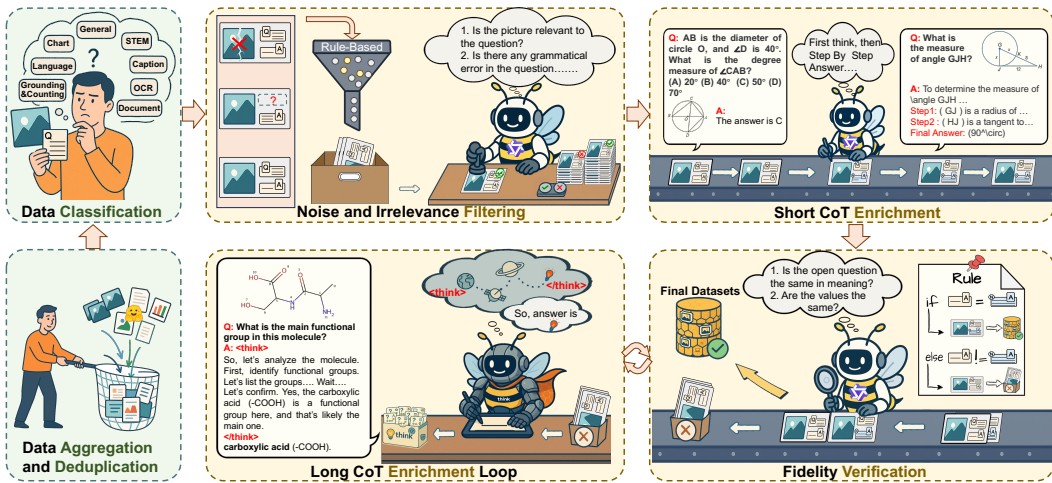

Figure 1: Overview of the HoneyPipe. After initial data preparation and filtering, the foundational path generates short CoT responses, which are then checked by Fidelity Verification. Complex instructions that fail this check are routed to the long CoT Enrichment loop. Here, a more powerful model generates a detailed long CoT response, which then undergoes the same Fidelity Verification. This dual-level architecture systematically builds the Honey-Data-15M.

- **Honey-Data-15M:** A dataset with 15M QA pairs, systematically cleaned of noise and enriched with a dual-level CoT reasoning, serving as a new cornerstone for the open community.
- **HoneyPipe:** The data curation pipeline and its underlying framework, DataStudio, offering the community a transparent and adaptable methodology that moves beyond static dataset releases.
- **Bee-8B:** An 8B model trained on Honey-Data-15M that achieves SOTA among fully open models and competes with semi-open counterparts, validating data quality and pipeline effectiveness.

## 2 HONEY-DATA-15M AND HONEYPIPE

To construct our Honey-Data-15M dataset, we introduce the data pipeline, HoneyPipe, as shown in Fig. 1. It is an automated and reproducible workflow designed to systematically address the twin challenges of widespread data noise and the gap in complex reasoning abilities identified as critical roadblocks for the open-source community. The entire pipeline is constructed from the modular components of our DataStudio. Its architecture features a nuanced, dual-level reasoning enrichment strategy, comprising a foundational path for large-scale short CoT enrichment and a specialized loop for generating long CoT responses to the most complex instructions. This structure allows us to transform a vast, raw data pool into a high-quality, dual-level CoT supervised fine-tuning (SFT) dataset. By offering the community a transparent and adaptable methodology, we move beyond static dataset releases. Each stage of this pipeline is detailed below.

### 2.1 STAGE 1: DATA AGGREGATION AND PREPARATION

The pipeline begins by assembling an initial pool of approximately 24 million image-text pairs from diverse community datasets, including LLaVA-OneVision (Li et al., 2025a), PixMo (Deitke et al., 2025), and MAmmoth-VL (Guo et al., 2025c), etc., where content overlap is a key challenge. To maximize data diversity and enhance processing efficiency, we performed rigorous deduplication at the pair level. Specifically, a sample was considered a duplicate and removed only if both its image (represented by a perceptual hash) and its textual instruction (represented by a simhash) were identical to those of another sample. This critical process significantly reduced redundancy, yielding a clean and unique set of image-instruction-response triplets. Finally, to guide subsequent processing, each sample was assigned a domain label (e.g., General, Chart, OCR, STEM). Instead of performing computationally expensive per-instance classification, we employed an efficient coarse-grained strategy at the data source level. Specifically, for each aggregated data source, we manually in-

spected approximately five representative image-text pairs to determine a single overarching topic label, which was then uniformly assigned to all samples originating from that source.

## 2.2 STAGE 2: NOISE AND IRRELEVANCE FILTERING

This stage is designed to purge the widespread data noise endemic to open-source datasets by integrating both rule-based and model-based filtering operators. The rule-based operators address specific formatting issues, such as removing samples with very small images, extreme aspect ratios, or repeated text in instructions. Orchestrated alongside these is a model-based filtering operator, which leverages the powerful Qwen2.5-VL-72B (Bai et al., 2025) model to ensure image-instruction consistency. This operator is prompted to assess whether an instruction is logical and answerable, and whether it is semantically related to the visual content. For example, it will flag an instruction such as "solve the function problem" as irrelevant to an image containing only oranges.

By integrating these different types of operators within this stage, we effectively pruned flawed samples, producing a clean set of image-instruction pairs ready for enrichment.

## 2.3 STAGE 3: SHORT CoT ENRICHMENT AND VERIFICATION

With a clean data foundation, this stage constitutes the foundational path of our dual-level CoT enrichment strategy. It targets instructions that require moderate reasoning by generating explicit, step-by-step CoT explanations. The process begins with data triage before proceeding to two main phases: enrichment and verification.

First, we identify data sources unsuitable for reasoning enhancement, such as samples from computer vision tasks like OCR or object detection. These samples bypass the enrichment process and are added to the final dataset. For all other samples, we proceed as follows:

**Short CoT Enrichment.** For samples slated for enrichment, we begin by preprocessing their instructions. We remove any head or tail prompts that might discourage detailed reasoning (e.g., "Answer directly") to elicit the model to produce a comprehensive, step-by-step response. We then use powerful open-source MLLMs (Qwen2.5-VL-72B/32B) to transform the simple, short-form responses into detailed reasoning paths. Notably, we do not add extra system prompts. This is because the models are already adept at generating multi-step responses, and a prompt could constrain their output diversity. This step is the primary source of the approximately 12.1 million short CoT samples in our dataset.

**Fidelity Verification.** To ensure the fidelity of the generated content, we employ a verification stage that operates on the "LLM-as-a-Judge" (Gu et al., 2024; Zheng et al., 2023). We use a verifier model (Qwen2.5-VL-72B) to perform a semantic comparison between the final conclusion of the newly generated CoT and the original response. The evaluation is twofold: for factual queries (objective questions), the final responses must match precisely; for descriptive or open-ended queries (subjective questions), thematic relevance and semantic consistency are required. Samples that pass this check are added to the final dataset. Samples that fail are not discarded but are instead routed to the long CoT enrichment loop for more specialized enrichment.

## 2.4 STAGE 4: LONG CoT ENRICHMENT LOOP

This loop constitutes the second level of the enrichment strategy, designed specifically for the most complex instructions that demand deep, multi-step problem-solving. This secondary path processes three primary types of inputs: 1) the samples that failed the fidelity verification in the previous stage Sec. 2.3; 2) samples from select data sources identified as inherently complex during our initial classification (e.g., VisualWebInstruct (Jia et al., 2025)), for which we proactively generate a long CoT version in addition to their short CoT counterpart; and 3) samples from datasets that have been validated in prior research (e.g., Vision-R1 (Huang et al., 2025)) as being particularly suitable for generating deep reasoning chains.

We leverage the top proprietary MLLMs to generate a more detailed solution. When tasked with the instructions, the model first generates a deep reasoning, often structured with tags like $<$ think $><$ /think $>$, before outputting the final response. This specialized reasoning level can handle complex instructions beyond the reach of the initial models.

Following this enrichment, each new long CoT response undergoes the same fidelity verification described in Sec. 2.3. This final validation ensures the correctness of the enriched response. The samples that successfully pass this check constitute the approximately 2.9 million long CoT data points in Honey-Data-15M. We discard any sample that fails this final verification, assuming it is erroneous, unsolvable, or too costly to annotate, even if top proprietary models cannot solve it.

## 2.5 HONEY-DATA-15M: HIGH QUALITY CORPUS WITH THE DUAL-LEVEL COT

The primary output of our pipeline is Honey-Data-15M, a large-scale, multimodal SFT dataset comprising 15 million meticulously curated samples. Our primary objective in developing Honey-Data-15M is to furnish the research community with a high-quality, reproducible resource that can serve as a new cornerstone for the fully open MLLM community. The compositional breakdown, illustrated in Fig. 2, reveals a diverse amalgamation of data sources, strategically chosen to cover a wide spectrum of domains and complexities. A more granular description of the source-specific statistics can be found in Fig. 3.

A defining feature of Honey-Data-15M is its enrichment with dual-level CoT reasoning, which forms the backbone of the dataset. We have integrated approximately 12.1 million short CoT samples, designed to instill foundational, step-by-step logical inference. Complementing these are 2.9 million long CoT samples, which challenge models with more intricate, multi-step reasoning problems that require a deeper synthesis of information. This dual-pronged approach ensures that models trained on Honey-Data-15M can develop both precision in concise reasoning and depth in complex problem-solving.

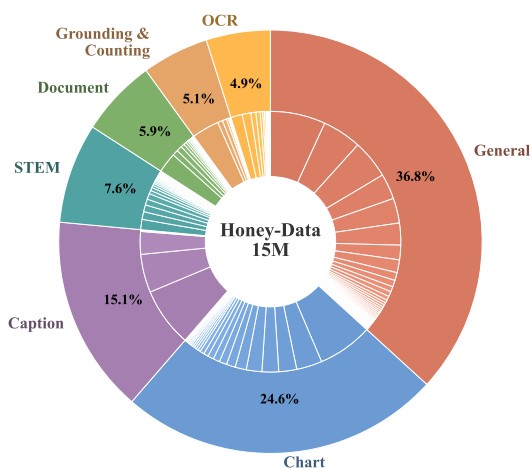

Figure 2: Distributions of Honey-Data-15M.

To forge powerful and well-rounded models, Honey-Data-15M applies its dual-level reasoning across a spectrum of critical domains, such as "General" for foundational visual understanding and "STEM" for symbolic reasoning, thereby ensuring comprehensive skill development.

# 3 BEE-8B TRAINING RECIPE

To validate the effectiveness of our Honey-Data-15M, we developed Bee-8B, an 8B multimodal large language model trained with a multi-stage recipe. This training process is intended to showcase the strengths of our Honey-Data-15M dataset, particularly its capacity to foster advanced complex reasoning.

## 3.1 MODEL ARCHITECTURE

Our model, Bee-8B, adopts a proven and effective MLLM architecture, drawing inspiration from leading open-source models like LLaVA-OneVision (Li et al., 2025a). It is built upon the powerful Qwen3-8B (Yang et al., 2025a) large language model, which serves as the foundation for reasoning and text generation. For visual understanding, we employ the open-source SigLIP2-so400m-patch14-384 (Tschannen et al., 2025) as our core vision encoder. To effectively handle images of varying resolutions and preserve fine-grained details, we use the Anyres strategy (Liu et al., 2024b). A simple two-layer MLP with a GELU activation function serves as the projector, mapping these aggregated visual features into the LLM's token embedding space.

**General (36.8%)**

| | | | | |
|---|---|---|---|---|
| SVIT-mix-665K(1019.1K)* | ALLaVA(721.1K)* | LLaVA-NeXT-Data(711.5K) | LLaVA-Instruct-300k(470.8K)* | Vision FLAN(458.8K)* |
| idefics375k(406.1K)* | SVIT-core-150K(269.4K)* | PixMo-CapQA(237.3K)* | LVIS-InstructV4(145.4K)* | ShareGPT4V(SAM)(122.5K)* | Cambrian (Filter)(99.6K)* |
| PixMo-AskModelAnything(96.3K) | ShareGPT4o(73.9K)* | NLVR2(54.1K) | GQA(45.7K)* | PixMo-Point-Explanations(44.9K) | LRV Normal(39.5K)* |
| HQ-Edit(36.4K) | ALFWorld(31.9K)* | Visual7W(30.7K)* | Co-Instruct(30.5K) | Cauldron(mulberry)(28.1K)* | A-OKVQA(27.0K)* |
| IconQA(26.5K) | VIST(21.1K) | KVQA(17.7K) | ContrastiveCaption(17.2K) | FlintstonesSV(17.0K) | InternVL-SA-1B-Caption(16.9K) |
| IDK(16.6K)* | COCO(16.3K)* | EST-VQA(13.5K) | Birds-to-Words(11.2K) | ART500K(11.0K) | DreamSim(11.0K) |
| ScanQA(10.2K) | MagicBrush(9.8K) | KonIQ-10k(9.4K) | Hateful Memes(7.3K)* | WebQA(6.6K) | nuScenes(6.5K) |
| Other(5.7K)* | Objects365(5.2K) | MMChat-Twitter-Post(4.3K) | NextQA(3.1K) | VSR(2.6K)* | New Yorker Caption(2.6K) |
| ViQuAE(2.4K)* | TQA(2.1K) | ShareGPT4V(Knowledge)(2.0K)* | WildVision(1.9K) | | |

**Chart (24.6%)**

| | | | | |
|---|---|---|---|---|
| TinyChart(1014.1K)* | DVQA(473.8K)* | UniChart(330.0K) | CoSyn(chart)(308.2K)* | ArxivQA(287.0K)* |
| FigureQA(206.8K)* | MMTab(165.6K) | PlotQA(138.7K)* | UReader QA(121.4K)* | RobuT WikiSQL(110.9K)* | CoSyn(table)(86.5K)* |
| CoSyn(diagram)(77.1K)* | TabMWP(46.3K)* | RobuT WTQ(44.2K)* | UReader KG(38.2K)* | Chart2Text(34.5K)* | ChartQA(34.4K)* |
| MMC-Instruction(27.1K) | RobuT SQA(26.6K)* | MAVIS-Function(25.3K) | CoSyn(graphic)(19.4K)* | VisText(15.6K)* | SciTSR(9.0K) |
| MultiHiertt(4.6K)* | SimChart9K(4.5K) | Other(3.9K)* | HiTab(3.6K)* | LRV Chart(3.3K)* | Infographic(2.2K)* |

**Caption (15.1%)**

| | | | | |
|---|---|---|---|---|
| COYO-Recaption(1091.8K)* | PixMo-Cap(706.9K) | WIT(419.9K)* | Sherlock(16.9K)* | ST-VQA(13.5K)* |
| Other(1.1K)* | | | | |

**STEM (7.6%)**

| | | | | |
|---|---|---|---|---|
| VisualWebInstruct(filtered)(190.4K)* | MapQA(142.2K)* | VisualWebInstruct(127.1K)* | MetaMathQA(110.0K) | Geo170K(106.6K)* |
| MAVIS-Metagen(62.5K)* | GeoQA+(60.3K)* | MAVIS-Geo(43.9K)* | CoSyn(math)(37.4K)* | AI2D(27.8K)* | PMC-VQA(25.1K)* |
| RAVEN(23.1K)* | PathVQA(21.8K)* | CoSyn(music)(21.2K)* | CoSyn(chemical)(18.2K)* | CoSyn(circuit)(17.0K)* | MathV360K(TQA)(14.2K)* |
| ScienceQA(12.6K)* | Geometry3K(11.4K)* | AI2D(InternVL)(10.6K)* | MMChem(10.0K) | WebSight(7.6K)* | UniGeo(7.6K)* |
| Other(7.4K)* | GeomVerse(6.5K)* | AI2D(GPT4V)(5.5K)* | VizWiz(5.3K)* | VQA-RAD(1.6K)* | CMM-Math(1.5K) |
| InterGPS(1.3K)* | | | | |

**Document (5.9%)**

| | | | | |
|---|---|---|---|---|
| Ureader Chart(371.0K)* | OCR-VQA(107.8K)* | CoSyn(document)(101.5K)* | ScreenQA(64.9K)* | FinTabNet(50.1K) |
| TextVQA(39.1K)* | EATEN(30.0K) | DocVQA(23.8K)* | LLaVAR GPT4(23.3K)* | Docmatix(19.1K) | CoSyn(nutrition)(13.0K)* |
| InfoVQA(9.8K)* | UreaderOCR(5.5K)* | Other(4.3K)* | DocReason(4.0K) | InfographicVQA(4.0K)* | VisualMRC(3.3K)* |
| POIE(2.2K) | | | | |

**Grounding & Counting (5.1%)**

| | | | | |
|---|---|---|---|---|
| CLEVR(504.2K)* | TallyQA(91.6K)* | VisualGenome(77.4K) | IconQA(35.9K)* | TQA(20.6K)* |
| MovieNet(7.4K) | CLEVR-Math(7.2K)* | Super-CLEVR(6.8K)* | MathV360K(VQA-AS)(4.1K)* | CLEVR-Change(3.0K) | Other(1.7K)* |

**OCR (4.9%)**

| | | | | |
|---|---|---|---|---|
| K12 Printing(211.1K) | ArXiv OCR(159.3K) | HME(93.6K) | VCR-Wiki(77.9K) | TextOCR(47.7K) |
| Sroie(31.2K) | ICDAR-LSVT-zh(28.2K) | ReCTs(17.0K) | CTW(13.6K) | Rendered Text(10.0K) | ICDAR2017(9.9K) |
| Chrome-Writing(9.2K) | MTWI(zh)(8.3K) | IAM(5.7K) | ICDAR2019(3.4K) | Orand-Car-A(2.0K) | IIIT 5K(2.0K) |

Figure 3: Data collection of Honey-Data-15M. A detailed breakdown of our dataset's composition across seven major categories. The number of samples (in thousands) is listed for each source. The * denotes that the data contains the long CoT response.

## 3.2 MODEL TRAINING

Our training methodology is a five-stage process, detailed in Tab. 1, which progressively builds the model's capabilities from basic perception to complex reasoning. The overarching goal is to first establish a robust vision-language foundation, then instill deep reasoning abilities using our Honey-Data-15M dataset, and finally polish the model's outputs for enhanced reliability.

Table 1: Detailed configuration for each training stage of Bee-8B.

| Stages | Stage 1 | Stage 2 | Stage 3 | Stage 4 | Stage 5 |
|---|---|---|---|---|---|
| Purpose | MLP Warmup | Vision-Language Alignment | Multimodal SFT | Efficient Refinement SFT | Policy Optimization RL (GRPO) |
| Batch size | 512 | 256 | 256 | 256 | 512 |
| Learning Rate | 1e-3 | 4e-5 | 5e-5 | 3e-5,5e-6 | 2e-6 |
| Dataset Items | 1M | 14M | 15M | 1M | 50K |
| Packed Sequence Length | 8192 | 16384 | 16384 | 16384 | - |
| Training Epochs | 1 | 1 | 1 | 1 | - |
| Trainable Components | MLP | All | All | All | All |

The training begins with two foundational stages. The first stage consists of an MLP warmup, where only the projector is trained to efficiently align the visual and language feature spaces without altering the pretrained backbones. This is followed by a full-parameter vision-language alignment stage, where the model is trained on a large corpus of image-text pairs and text-only data to build core multimodal capabilities while preserving the LLM's intrinsic cognitive abilities.

The subsequent three stages focus on advanced instruction tuning and refinement. The pivotal third stage is a large-scale SFT on the entire Honey-Data-15M dataset, designed to instill the complex reasoning patterns from our dual-level CoT data. The fourth stage is an efficient refinement SFT on Honey-Data-1M, a subset of Honey-Data-15M curated for a more rational topic distribution, thereby further polishing the model's capabilities. Finally, the fifth stage employs Group Relative Policy Optimization (GRPO) to improve response quality and reliability by mitigating common generation issues like text repetition. The efficacy of this final optimization phase hinges on the high-quality model produced by the SFT stages, which in turn validates our curated data. Further details on the data composition, sources, and specific configurations for each stage are provided in Appx. F.

## 4 EXPERIMENTS

To validate the effectiveness of our Honey-Data-15M, we conduct a comprehensive experimental evaluation of Bee-8B, beginning with our evaluation setup (Sec. 4.1). We then benchmark Bee-8B against leading fully open and semi-open MLLMs to demonstrate its capabilities (Sec. 4.2) and conclude with ablation studies to validate the high quality of our Honey-Data-15M and quantify the impact of our data curation strategy (Sec. 4.3).

### 4.1 EVALUATION SETUP

We evaluated our model using a customized VLMEvalKit (Duan et al., 2024), which we adapted to enable LLM-based judging on benchmarks like DocVQA. Our model was evaluated with the thinking mode and a maximum response length of 16,384 tokens. For LLM-based judging, we employed Qwen3-32B (Yang et al., 2025a) in a non-thinking mode. Additional details are provided in the Appx. H.

### 4.2 EVALUATION OF MODEL CAPABILITY

As shown in the comprehensive benchmark results in Tab. 2, Bee-8B achieves across-the-board performance improvements over existing fully open models (Li et al., 2025a; Deitke et al., 2025) and stands as a highly competitive alternative to recent semi-open models (Bai et al., 2025; Team et al., 2025c; Wang et al., 2025b). Its most significant advantages are observed in factual accuracy and complex multi-step reasoning, directly reflecting the strengths of our Honey-Data-15M dataset. More results for all benchmarks are available in Appx. A.1. Key results are detailed as follows:

**General VQA Tasks:** Bee-8B showcases a robust and well-rounded performance across a wide array of general visual question answering tasks. On comprehensive, multi-domain benchmarks such as MMMU (Yue et al., 2024) and MMStar (Chen et al., 2024c), it achieves highly competitive scores of 66.8 and 71.4, respectively, demonstrating its strong general knowledge base. The model's superior performance becomes particularly apparent on benchmarks that test specific VQA capabilities. For example, on MMMU-Pro (Yue et al., 2025a), a benchmark for professional-level knowledge, Bee-8B obtains a top score of 50.7. This establishes a commanding lead of 3.6% over the next-best competitor, Qwen2.5-VL-7B (Bai et al., 2025), underscoring its advanced cognitive abilities. Similarly, on CountBench (Paiss et al., 2023), it ranks first with an exceptional score of 93.0. This leading performance extends to other tasks, including a top score of 83.9 on the MMVet (Yu et al., 2024b). Furthermore, it demonstrates superior real-world knowledge by securing the top rank on RealWorldQA (xAI, 2024) with a score of 73.1. This suite of results confirms that Bee-8B not only possesses a broad understanding of general multimodal information but also excels in a variety of core visual skills.

**Document, Chart, and OCR Tasks:** In tasks involving structured visual content like documents, tables, and charts, Bee-8B demonstrates strong performance. Its skill in chart analysis is evident on ChartQA (Masry et al., 2022), where it achieves a highly competitive score of 86.7, confirming its robust data parsing abilities. This proficiency is most pronounced in the challenging area of scientific document analysis. On the CharXiv benchmark (Wang et al., 2024c), Bee-8B secures the top rank in both descriptive questions (DQ) and reasoning questions (RQ) with scores of 84.8 and 57.3, respectively. For the reasoning task, its score establishes an impressive lead of nearly 12% over the closest competitor, Keye-VL (45.4) (Team et al., 2025c), underscoring its advanced capacity for deep semantic inference. These results validate Bee-8B's powerful abilities to not only extract precise information but also to deeply comprehend the context of structured visual data.

**Math and Reasoning Tasks:** The most significant advancements delivered by Bee-8B are in complex math and reasoning, directly validating the effectiveness of our CoT-enriched Honey-Data-15M dataset. It consistently delivers exceptional performance across a suite of benchmarks designed to test quantitative and logical problem-solving. The model's leadership is most evident on Math-Verse (Zhang et al., 2024b), where the RL-tuned version scores a top-ranking 67.0. This represents a clear improvement of 5.5% over the strong semi-open model InternVL3.5-8B (Wang et al., 2025b), showcasing its superior visual-mathematical skills. Furthermore, on LogicVista (Xiao et al., 2024), Bee-8B also achieves the top score of 61.3, surpassing the next-best competitor by 4% and demonstrating its versatile reasoning abilities. Its robustness is further confirmed on DynaMath (Zou et al.,

Table 2: Evaluation of Bee-8B against other MLLMs. We distinguish between fully open (*) and semi-open (†) models. The **top** and **second-best** scores for each benchmark are highlighted.

| Task | Benchmark | LLaVA OneVision-7B* | Molmo -7B-D* | Qwen2.5 -VL-7B† | Keye-VL -8B† | InternVL3.5 -8B† | Bee-8B -SFT* | Bee-8B -RL* |
|---|---|---|---|---|---|---|---|---|
| **General VQA** | AI2D | 81.4 | 81.0 | 84.3 | **86.7** | 84.0 | 83.8 | **85.3** |
| | BLINK$_{val}$ | 48.2 | 49.7 | **56.4** | 52.0 | **59.5** | 52.5 | 55.0 |
| | CountBench | – | 84.8 | 74.1 | 78.0 | – | **90.5** | **93.0** |
| | HallusionBench$_{avg}$ | 31.6 | 46.4 | 52.9 | **67.0** | 54.5 | **59.8** | 58.2 |
| | MMBench-CN$_{dev}$ | – | – | 81.3 | **92.0** | – | 81.2 | **84.2** |
| | MMBench-EN$_{dev}$ | 80.8 | – | 82.1 | **91.5** | – | 83.0 | **85.5** |
| | MMMU$_{val}$ | 48.8 | 45.3 | 58.6 | **71.4** | **73.4** | 66.8 | 66.1 |
| | MMMU-Pro$_{standard}$ | 29.5 | – | 34.7 | 47.1 | – | **50.4** | **50.7** |
| | MMStar | 61.7 | 56.1 | 63.9 | **75.5** | 69.3 | 69.0 | **71.4** |
| | MMT-Bench$_{val}$ | 59.3 | 56.3 | 63.6 | 65.9 | **66.7** | 64.6 | **67.0** |
| | MMVet | 57.5 | 41.5 | 67.1 | 79.0 | 83.1 | **83.3** | **83.9** |
| | MMVP | – | – | 73.3 | 79.0 | – | **80.7** | **82.0** |
| | POPE$_{avg}$ | 88.4 | **89.0** | 86.4 | 86.0 | **88.7** | 84.0 | 84.8 |
| | RealWorldQA | 66.3 | **70.7** | 68.5 | 67.7 | 67.5 | 70.1 | **73.1** |
| | VisuLogic | – | – | 20.0 | **25.6** | – | 24.4 | **26.5** |
| | VLMs are Blind | 39.2 | – | 37.4 | **57.1** | – | 55.8 | **56.5** |
| **Table & Chart & OCR** | CharXiv$_{DQ}$ | – | – | 73.9 | 77.7 | 72.2 | **84.7** | **84.8** |
| | CharXiv$_{RQ}$ | – | – | 42.5 | 45.4 | 44.4 | **55.3** | **57.3** |
| | ChartQA$_{test}$ | 80.0 | 84.1 | **87.3** | 86.3 | 86.7 | **86.7** | 86.1 |
| | DocVQA$_{val}$ | – | – | **95.5** | **88.5** | – | 87.2 | 87.0 |
| | InfoVQA$_{val}$ | – | – | **81.4** | 67.4 | – | 72.3 | **72.9** |
| | OCRBench | 62.2 | 65.6 | **86.4** | **85.1** | 84.0 | 83.1 | 82.5 |
| | SEED-Bench2-Plus | 65.4 | 67.6 | **70.4** | 69.4 | **70.8** | 67.7 | 68.5 |
| **Math & Reasoning** | DynaMath$_{worst}$ | 9.0 | – | 21.0 | 37.3 | 37.7 | **41.3** | **40.5** |
| | LogicVista | 33.3 | – | 44.1 | 54.8 | **57.3** | 56.8 | **61.3** |
| | MathVerse$_{vision\_only}$ | 26.2 | 4.2 | 25.1 | 59.8 | 61.5 | **61.9** | **67.0** |
| | MathVision | 18.3 | 16.2 | 25.4 | 46.0 | **56.8** | 46.8 | **50.0** |
| | MathVista$_{mini}$ | 63.2 | 51.6 | 68.2 | **80.7** | 78.4 | 78.6 | **81.4** |
| | WeMath | 20.9 | – | 35.2 | **60.7** | 57.0 | 55.0 | **59.8** |

2025), where it again secures the top score of 41.3. This consistent pattern of leading performance on challenging reasoning tasks confirms that our high-quality data strategy has successfully instilled the model with complex reasoning capabilities.

In summary, Bee-8B establishes a new performance bar for fully open models, particularly in factual accuracy and complex reasoning, and proves highly competitive with recent semi-open models. These findings confirm our core thesis: a focus on high-quality data curation is critical for creating models that can rival leading semi-open models.

## 4.3 ABLATION STUDY

**Ablation for HoneyPipe:** To precisely isolate the effects of our data curation pipeline, we conducted an ablation study with three data subsets, visualized in Fig. 4. We established a baseline with $D_{raw}$ (1.2M raw samples). Our main curated set is $D_{curated}$ (960K samples after full filtering and short CoT enrichment). To isolate the impact of the enriched responses themselves, we created $D_{no-CoT}$, an identical 960K set to $D_{curated}$, but with the new CoT responses replaced by the original, simple responses. The results reveal a clear hierarchy ($D_{curated} > D_{no-CoT} > D_{raw}$). The significant improvement from $D_{raw}$ to $D_{no-CoT}$ demonstrates the large combined benefit of noise filtering and our data selection process. More importantly, the performance leap from $D_{no-CoT}$ to $D_{curated}$ clearly demonstrates the direct impact of the CoT enrichment itself. The superior performance, especially in reasoning-heavy benchmarks like MathVista and CharXiv-RQ, is therefore directly attributable to training on detailed, step-by-step rationales. This study provides powerful evidence that data cleaning and CoT enrichment are critical.

**Ablation for Honey-Data-1M:** To validate the effectiveness of our curated 1M data subset, we conducted an ablation study with results visualized in Fig. 5. We compare the original Qwen2.5-VL-7B base model against two variants fine-tuned from our Stage 2 checkpoint: one trained on Random-1M (a randomly sampled 1M subset) and another on Honey-Data-1M (our meticulously curated 1M subset). The results clearly demonstrate the superiority of our selection strategy, as the model fine-tuned on Honey-Data-1M outperforms the variant trained on the randomly sampled Random-1M. Notably, fine-tuning on just this 1M curated subset was sufficient for our model to surpass the

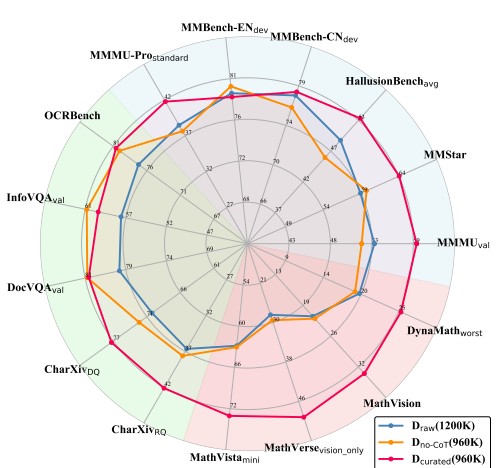

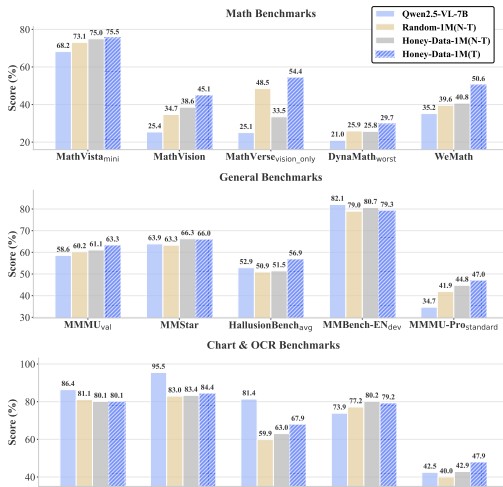

Figure 4: Visual comparison of model performance in our ablation study. The radar chart illustrates the step-by-step impact of our curation pipeline. The performance lift from $D_{\text{raw}}$ (baseline) to $D_{\text{no-CoT}}$ shows the benefit of noise filtering, while the larger leap to $D_{\text{curated}}$ highlights the critical contribution of Chain-of-Thought enrichment, especially in reasoning-heavy domains.

Figure 5: Performance comparison of models fine-tuned on different 1M data subsets. The model trained on our curated Honey-Data-1M consistently and outperforms both the Qwen2.5-VL-7B base model and the model trained on a Random-1M subset, highlighting the efficiency of our data selection strategy.

original Qwen2.5-VL-7B on nearly half of the evaluated benchmarks. This study provides powerful evidence for both the high quality of our data and the effectiveness of our selection strategy in cultivating advanced reasoning skills.

## 5 RELATED WORK

Multimodal Large Language Models (MLLMs) have demonstrated immense potential across diverse real-world scenarios (Li et al., 2026; Xu et al., 2026; Ning et al., 2026; Tang et al., 2025a; Wei et al., 2025; Tang et al., 2025b). The modern wave of MLLMs began with GPT-4V (OpenAI, 2023), which showed that a general LLM (OpenAI, 2023; Touvron et al., 2023; Bai et al., 2023a) could process visual inputs. In the open-source community, LLaVA (Liu et al., 2023c) established a standard paradigm of a frozen vision encoder (Radford et al., 2021; Tschannen et al., 2025; Zhai et al., 2023), a lightweight connector, and visual instruction tuning, proving that GPT-generated instructions were sufficient for building a practical multimodal assistant. Subsequently, the field's focus shifted to data, which has produced a significant data gap. Proprietary models (OpenAI, 2025; Comanici et al., 2025; Guo et al., 2025b) leverage massive, ever-evolving private datasets. Semi-open models (Wang et al., 2024b; Yao et al., 2024; Zhang et al., 2025a; Bai et al., 2025; 2023b; Yue et al., 2025b; Team et al., 2025c;b; Yang et al., 2025b; Team et al., 2025a) retain an edge through private data curation, even with released weights. In contrast, fully open models (Liu et al., 2024b; Wang et al., 2025c; Guo et al., 2025c; Deitke et al., 2025; Li et al., 2025a) are constrained in both the scale and quality of their public data. This performance hierarchy stems more from data access and pipelines than from architectural tweaks. Accordingly, our work centers not on novel architectures but on a high-quality, reproducible data curation pipeline designed to narrow this gap for fully open models at a reasonable cost.

## 6 AVAILABILITY

We are committed to open science and the reproducibility of our results. We will publicly release the full suite of resources developed in this work, including:

- **Dataset:** The complete Honey-Data-15M corpus.
- **Model:** The weights for the final Bee-8B model.
- **Codebase:** The source code for our data curation pipeline (HoneyPipe) and the underlying framework (DataStudio). And all scripts, configurations, and the code required to train and evaluate.
- **Training Resources:** To support detailed analysis and community research, we will also release the intermediate model checkpoints for each stage of our training recipe, along with the specific training data subsets used for each stage.

## 7 CONCLUSION

In this work, we addressed the critical data quality gap that hinders the progress of open-source MLLMs. We introduced Honey-Data-15M, a 15M-sample SFT dataset systematically cleaned of noise and enriched with a dual-level Chain-of-Thought (CoT) reasoning structure, all constructed using our open-source, model-driven pipeline, HoneyPipe. To validate our approach, we trained Bee-8B, which establishes a new state-of-the-art among fully open models and proves highly competitive with several leading semi-open models. Our experiments reveal standout performance in complex math and reasoning, a direct outcome of our targeted CoT enrichment, with ablation studies confirming the significant impact of our curation process. Ultimately, our work suggests a clear path forward for the community: prioritizing data quality through transparent, reproducible methods is a more effective strategy than competing on data volume. The full-stack suite of our dataset, pipeline, and model aims to facilitate this data-centric approach and empower further innovation within the open-source ecosystem. By open-sourcing not just our dataset but the entire curation methodology, we hope to provide a cornerstone for a new wave of collaborative and competitive open-source MLLM development.

**Acknowledgement.** This work was supported by Fundamental and Interdisciplinary Disciplines Breakthrough Plan of the Ministry of Education of China (No. JYB2025XDXM101), the National Natural Science Foundation of China (project No. 62495060, 623B2057), and the Research Grant of Tsinghua-Tencent Joint Laboratory for Internet Innovation Technology.

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

# APPENDIX

## A  ADDITIONAL EXPERIMENTS

In this section, we provide a series of additional experiments to offer a more granular understanding of our model's performance and the key components of our training methodology. We investigate the specific contributions of different training stages and analyze the model's behavior under various inference conditions. These analyses serve to validate our design choices and highlight the critical factors that enable the model's advanced reasoning capabilities.

### A.1  ABLATION OF DIFFERENT REASONING MODES AND STAGES

To further analyze our model's performance and the impact of our training methodology, we conducted a comprehensive ablation study comparing the model at three key checkpoints: the initial SFT model (Stage 3), the model after refinement SFT (Stage 4), and the final model after reinforcement learning (Stage 5). For the SFT models, we also assessed performance across both short CoT (N-T) and long CoT (T) inference modes. This multifaceted analysis aims to understand the progressive impact of each training stage and the model's behavior across different reasoning complexities.

As detailed in Tab. 3 to 6, we assessed performance across a wide array of benchmarks in two distinct inference modes:

Table 3: Performance comparison of our model after Stage 3, Stage 4, and Stage 5 on general VQA benchmarks (Part 1). The **top** and **second-best** scores for each benchmark are highlighted.

| Model | Mode | AI2D | BLINK val | Count Bench | Hallusion Bench$_{avg}$ | MMBench -CN$_{dev}$ | MMBench -EN$_{dev}$ | MMMU val | MMMU -Pro$_{standard}$ |
|---|---|---|---|---|---|---|---|---|---|
| **Stage3** | **N-T** | 82.7 | 53.3 | **93.2** | 54.2 | 79.6 | 81.3 | 58.9 | 43.2 |
| **Stage3** | **T** | 83.7 | 52.6 | 91.4 | 57.0 | **83.0** | 81.9 | **66.7** | 48.5 |
| **Stage4** | **N-T** | 81.2 | **54.9** | 92.4 | 56.8 | 80.7 | 82.7 | 59.8 | 43.9 |
| **Stage4** | **T** | **83.8** | 52.5 | 90.5 | **59.8** | 81.2 | **83.0** | **66.8** | **50.4** |
| **Stage5** | **T** | **85.3** | **55.0** | **93.0** | **58.2** | **84.2** | **85.5** | 66.1 | **50.7** |

**Short CoT (N-T) mode**: We prompt the model with `<think>\n\n</think>` to elicit a concise reasoning path with fewer steps (short CoT).

**Long CoT (T) mode**: We use the prompt `<think>\n` to encourage the model to produce a more detailed and in-depth reasoning process (long CoT).

The model exhibits strong performance across both inference modes, reflecting an effective alignment between its vision module and the dual-level reasoning capabilities inherent in the Qwen3 LLM. The results show a clear progression of improvements. First, the Stage 4 refinement yields

Table 4: Performance comparison of our model after Stage 3, Stage 4, and Stage 5 on general VQA benchmarks (Part 2). The **top** and **second-best** scores for each benchmark are highlighted.

| Model | Mode | MMStar | MMT -Bench$_{val}$ | MMVet | MMVP | POPE avg | RealWorldQA | VisuLogic | VLMs are Blind |
|-------|------|--------|--------------------|-------|-------|----------|-------------|-----------|----------------|
| Stage3 | N-T | 66.5 | 62.3 | 69.3 | **81.0** | **88.0** | 70.7 | **26.1** | **56.9** |
| Stage3 | T | 68.0 | 63.8 | **84.1** | 80.0 | 84.4 | **72.5** | 24.0 | 53.9 |
| Stage4 | N-T | 67.5 | 63.2 | 69.2 | 79.0 | **86.2** | 70.5 | 24.2 | 52.9 |
| Stage4 | T | **69.0** | **64.6** | 83.3 | 80.7 | 84.0 | 70.1 | 24.4 | 55.8 |
| Stage5 | T | **71.4** | **67.0** | **83.9** | **82.0** | 84.8 | **73.1** | **26.5** | **56.5** |

Table 5: Performance comparison on benchmarks for table, chart, and document understanding. The **top** and **second-best** scores for each benchmark are highlighted.

| Model | Mode | CharXiv DQ | CharXiv RQ | ChartQA test | DocVQA val | InfoVQA val | OCRBench | SEED -Bench2-Plus |
|-------|------|------------|------------|--------------|------------|-------------|----------|-------------------|
| Stage3 | N-T | 81.7 | 47.0 | 82.1 | 86.3 | 66.5 | **84.2** | 66.7 |
| Stage3 | T | 83.0 | 53.4 | **86.8** | **87.3** | **72.5** | **84.1** | 67.7 |
| Stage4 | N-T | 84.4 | 48.3 | 79.4 | 86.3 | 66.7 | 83.0 | 67.2 |
| Stage4 | T | **84.7** | **55.3** | **86.7** | **87.2** | 72.3 | 83.1 | **67.7** |
| Stage5 | T | **84.8** | **57.3** | 86.1 | 87.0 | **72.9** | 82.5 | **68.5** |

discernible gains, which we attribute to the high quality of the curated 1M subset. Subsequently, the Stage 5 reinforcement learning provides a further significant boost. This stage primarily enhances the model's reliability by mitigating common generation issues such as response repetition, which in turn elevates its final performance on the majority of benchmarks, particularly in complex math and reasoning. This finding underscores that our multi-stage recipe, combining high-quality data curation with final policy optimization, is critical for unlocking a model's most advanced reasoning abilities.

## A.2 SENSITIVITY ANALYSIS OF JUDGE MODELS

As our primary evaluation pipeline utilizes a Qwen-based judge (Qwen3-32B) to assess a Qwen-based model (Bee-8B), there is a potential risk of "self-preference" bias. To rigorously quantify the sensitivity of our results to the choice of the evaluator, we conducted a cross-family evaluation using **GLM-4.5-FP8** (Zeng et al., 2025) as an independent judge across all 29 benchmarks.

Tab. 7 presents a comprehensive comparison of the performance of both Bee-8B-SFT and Bee-8B-RL when evaluated by the original Qwen judge versus the GLM judge. The results indicate a high degree of consistency between the two evaluators.

While we observe discrepancies in specific benchmarks, most notably in ChartQA, DocVQA, and InfoVQA, where the GLM judge tends to be stricter, the overall impact on the global average is minimal. For Bee-8B-RL, the Global Average score shifts only slightly by -0.3 points (from 70.2 with the Qwen judge to 69.9 with the GLM judge). This minimal deviation confirms that while same-family preference may exist to a minor degree or judges may have different calibration thresholds

Table 6: Performance comparison on mathematical and logical reasoning benchmarks. The **top** and **second-best** scores for each benchmark are highlighted.

| Model | Mode | DynaMath worst | LogicVista | MathVerse vision_only | MathVision | MathVista mini | WeMath |
|-------|------|----------------|------------|-----------------------|------------|----------------|--------|
| Stage3 | N-T | 34.3 | 52.6 | 61.9 | 37.8 | 78.1 | 43.1 |
| Stage3 | T | 35.9 | 54.1 | **63.3** | 42.8 | **78.8** | 52.7 |
| Stage4 | N-T | 31.1 | 52.1 | 62.9 | 40.6 | 78.6 | 45.5 |
| Stage4 | T | **41.3** | **56.8** | 61.9 | **46.8** | 78.6 | **55.0** |
| Stage5 | T | **40.5** | **61.3** | **67.0** | **50.0** | **81.4** | **59.8** |

Table 7: Sensitivity analysis of Bee-8B performance using different LLM judges (Qwen3-32B vs. GLM-4.5-FP8). We report the scores for both SFT and RL stages. The **Average** row shows the mean score across all 29 benchmarks. The minimal delta in the global average confirms the robustness of our evaluation.

| Task | Benchmark | Bee-8B-SFT | | | Bee-8B-RL | | |
|---|---|---|---|---|---|---|---|
| | | Qwen Judge | GLM Judge | Δ | Qwen Judge | GLM Judge | Δ |
| **General VQA** | AI2D | 83.8 | 83.7 | -0.1 | 85.3 | 85.3 | 0.0 |
| | $BLINK_{val}$ | 52.5 | 52.7 | +0.2 | 55.0 | 55.2 | +0.2 |
| | CountBench | 90.6 | 90.3 | -0.3 | 93.0 | 92.8 | -0.2 |
| | $HallusionBench_{avg}$ | 59.8 | 59.4 | -0.4 | 58.2 | 58.6 | +0.4 |
| | $MMBench-CN_{dev}$ | 81.2 | 81.2 | 0.0 | 84.2 | 84.2 | 0.0 |
| | $MMBench-EN_{dev}$ | 83.0 | 83.0 | 0.0 | 85.5 | 85.7 | +0.2 |
| | $MMMU_{val}$ | 66.8 | 66.0 | -0.8 | 66.1 | 65.6 | -0.5 |
| | $MMMU-Pro_{standard}$ | 50.5 | 49.0 | -1.5 | 50.7 | 50.3 | -0.4 |
| | MMStar | 69.0 | 68.9 | -0.1 | 71.4 | 71.4 | 0.0 |
| | $MMT-Bench_{val}$ | 64.6 | 64.6 | 0.0 | 67.0 | 67.1 | +0.1 |
| | MMVet | 83.3 | 85.0 | +1.7 | 83.9 | 83.7 | -0.2 |
| | MMVP | 80.7 | 80.0 | -0.7 | 82.0 | 82.0 | 0.0 |
| | $POPE_{avg}$ | 84.0 | 84.0 | 0.0 | 84.8 | 84.8 | 0.0 |
| | RealWorldQA | 70.1 | 70.1 | 0.0 | 73.1 | 72.8 | -0.3 |
| | VisuLogic | 24.4 | 24.4 | 0.0 | 26.5 | 26.5 | 0.0 |
| | VLMs are Blind | 55.8 | 55.8 | 0.0 | 56.5 | 56.5 | 0.0 |
| **Table & Chart & OCR** | $CharXiv_{DQ}$ | 84.7 | 83.6 | -1.1 | 84.8 | 84.8 | 0.0 |
| | $CharXiv_{RQ}$ | 55.3 | 54.0 | -1.3 | 57.3 | 55.8 | -1.5 |
| | $ChartQA_{test}$ | 86.7 | 84.2 | -2.5 | 86.1 | 83.8 | -2.3 |
| | $DocVQA_{val}$ | 87.2 | 83.6 | -3.6 | 87.0 | 83.4 | -3.6 |
| | $InfoVQA_{val}$ | 72.3 | 69.9 | -2.4 | 72.9 | 70.7 | -2.2 |
| | OCRBench | 83.1 | 83.1 | 0.0 | 82.5 | 82.5 | 0.0 |
| | SEED-Bench2-Plus | 67.7 | 67.8 | +0.1 | 68.5 | 68.5 | 0.0 |
| **Math & Reasoning** | $DynaMath_{worst}$ | 41.3 | 41.5 | +0.2 | 40.5 | 40.5 | 0.0 |
| | LogicVista | 56.8 | 56.2 | -0.6 | 61.3 | 60.9 | -0.4 |
| | $MathVerse_{vision\_only}$ | 61.9 | 61.5 | -0.4 | 67.0 | 66.9 | -0.1 |
| | MathVision | 46.8 | 47.6 | +0.8 | 50.0 | 50.2 | +0.2 |
| | $MathVista_{mini}$ | 78.6 | 78.7 | +0.1 | 81.4 | 81.4 | 0.0 |
| | WeMath | 55.0 | 54.8 | -0.2 | 59.8 | 60.0 | +0.2 |
| **Average** | **All Benchmarks** | **68.6** | **68.2** | **-0.4** | **70.2** | **69.9** | **-0.3** |

for specific tasks, our model's strong performance is fundamentally robust and not an artifact of a specific evaluator.

### A.3 EVALUATION STABILITY ANALYSIS

To verify the stability of our results, we repeated the evaluation for both Bee-8B-SFT and Bee-8B-RL across 5 independent inference runs. We report the scores for each individual run, the calculated mean, and the standard deviation (Std) in Tab. 8. We also provide the average score across all 29 benchmarks for each run. The extremely low standard deviation values across all benchmarks and the stable global average demonstrate that the model's performance is highly consistent and reproducible.

### A.4 HUMAN EVALUATION OF DATA QUALITY

To verify that our automated curation aligns with human standards, we conducted a rigorous blind evaluation. We recruited 14 valid evaluators to perform 532 pairwise comparisons between *Original* and *Enriched* responses across 38 randomly sampled questions. The assessment covered four key dimensions: **Accuracy** (factual correctness), **Reasoning & Explanation** (logical depth), **Instruction Following** (constraint adherence), and **Expression Style** (fluency and structure).

The results, summarized in Tab. 9, reveal a strong alignment with human preferences. The dominant Tie rate (83.65%) in Accuracy validates that our Fidelity Verification module effectively maintains

Table 8: Robustness analysis of Bee-8B (SFT and RL) across 5 independent runs. The table reports the score for each run, the mean, and the standard deviation.

| Task | Benchmark | Bee-8B-SFT (5 Runs) | | | | | | | Bee-8B-RL (5 Runs) | | | | | | |
|---|---|---|---|---|---|---|---|---|---|---|---|---|---|---|---|
| | | R1 | R2 | R3 | R4 | R5 | Mean | Std | R1 | R2 | R3 | R4 | R5 | Mean | Std |
| General VQA | AI2D | 83.8 | 83.5 | 83.7 | 84.1 | 84.0 | 83.8 | 0.2 | 85.3 | 85.1 | 85.7 | 84.8 | 84.5 | 85.1 | 0.4 |
| | BLINK$_{val}$ | 52.5 | 53.1 | 51.8 | 51.8 | 50.3 | 51.9 | 1.0 | 55.0 | 55.1 | 56.1 | 55.0 | 53.2 | 54.9 | 1.0 |
| | CountBench | 90.6 | 92.2 | 91.0 | 92.2 | 91.4 | 91.5 | 0.7 | 93.0 | 92.2 | 91.8 | 93.8 | 93.4 | 92.9 | 0.9 |
| | HallusionBench$_{avg}$ | 59.8 | 57.9 | 59.0 | 59.3 | 59.9 | 59.2 | 0.8 | 58.2 | 58.5 | 60.6 | 59.7 | 60.9 | 59.6 | 1.2 |
| | MMBench-CN$_{dev}$ | 81.2 | 82.2 | 81.8 | 82.0 | 81.7 | 81.8 | 0.4 | 84.2 | 83.8 | 84.4 | 83.8 | 84.1 | 84.1 | 0.3 |
| | MMBench-EN$_{dev}$ | 83.0 | 82.6 | 82.7 | 83.4 | 83.9 | 83.1 | 0.6 | 85.5 | 84.6 | 85.3 | 84.6 | 84.9 | 84.9 | 0.5 |
| | MMMU$_{val}$ | 66.8 | 65.3 | 65.4 | 65.7 | 64.8 | 65.6 | 0.7 | 66.1 | 68.0 | 67.7 | 67.9 | 67.1 | 67.4 | 0.8 |
| | MMMU-Pro$_{standard}$ | 50.5 | 48.8 | 50.0 | 50.0 | 48.7 | 49.6 | 0.8 | 50.7 | 52.1 | 49.7 | 50.8 | 50.7 | 50.8 | 0.8 |
| | MMStar | 69.0 | 68.7 | 70.3 | 68.5 | 69.4 | 69.2 | 0.7 | 71.4 | 72.9 | 70.3 | 70.9 | 71.9 | 71.5 | 1.0 |
| | MMT-Bench$_{val}$ | 64.6 | 65.0 | 65.2 | 64.6 | 65.6 | 65.0 | 0.4 | 67.0 | 66.6 | 67.3 | 67.5 | 67.2 | 67.1 | 0.3 |
| | MMVet | 83.3 | 81.7 | 82.6 | 78.9 | 82.8 | 81.8 | 1.8 | 83.9 | 85.9 | 83.6 | 82.1 | 82.2 | 83.5 | 1.5 |
| | MMVP | 80.7 | 82.7 | 80.7 | 79.3 | 79.3 | 80.5 | 1.4 | 82.0 | 80.7 | 83.0 | 84.0 | 79.7 | 81.9 | 1.7 |
| | POPE$_{avg}$ | 84.0 | 83.3 | 84.2 | 83.3 | 84.1 | 83.8 | 0.5 | 84.8 | 85.0 | 84.7 | 84.9 | 84.4 | 84.8 | 0.2 |
| | RealWorldQA | 70.1 | 68.6 | 70.8 | 70.2 | 69.9 | 69.9 | 0.8 | 73.1 | 71.0 | 70.8 | 73.6 | 72.9 | 72.3 | 1.3 |
| | VisuLogic | 24.4 | 25.4 | 23.7 | 22.7 | 24.8 | 24.2 | 1.0 | 26.5 | 25.4 | 25.3 | 26.1 | 26.6 | 26.0 | 0.6 |
| | VLMs are Blind | 55.8 | 53.7 | 54.0 | 54.5 | 54.5 | 54.5 | 0.8 | 56.5 | 57.3 | 57.5 | 56.6 | 56.6 | 56.9 | 0.5 |
| Table & Chart & OCR | CharXiv$_{DQ}$ | 84.7 | 84.1 | 84.6 | 84.2 | 84.9 | 84.5 | 0.4 | 84.8 | 84.7 | 84.8 | 84.8 | 84.8 | 84.8 | 0.1 |
| | CharXiv$_{RQ}$ | 55.3 | 56.4 | 53.4 | 56.3 | 55.1 | 55.3 | 1.2 | 57.3 | 57.2 | 54.6 | 55.8 | 56.6 | 56.3 | 1.1 |
| | ChartQA$_{test}$ | 86.7 | 86.6 | 87.4 | 86.9 | 86.5 | 86.8 | 0.3 | 86.1 | 87.3 | 87.0 | 87.1 | 87.4 | 87.0 | 0.5 |
| | DocVQA$_{val}$ | 87.2 | 87.2 | 87.5 | 87.7 | 87.2 | 87.4 | 0.3 | 87.0 | 87.2 | 87.0 | 87.3 | 87.0 | 87.1 | 0.2 |
| | InfoVQA$_{val}$ | 72.3 | 71.8 | 72.1 | 72.2 | 72.3 | 72.1 | 0.2 | 72.9 | 73.5 | 72.8 | 72.5 | 72.5 | 72.9 | 0.4 |
| | OCRBench | 83.1 | 82.9 | 83.5 | 83.5 | 82.8 | 83.2 | 0.3 | 82.5 | 83.3 | 83.6 | 82.4 | 83.7 | 83.1 | 0.6 |
| | SEED-Bench2-Plus | 67.7 | 67.3 | 67.7 | 67.2 | 68.4 | 67.7 | 0.5 | 68.5 | 68.6 | 69.1 | 68.3 | 68.8 | 68.7 | 0.3 |
| Math & Reasoning | DynaMath$_{worst}$ | 41.3 | 40.3 | 39.9 | 38.7 | 40.1 | 40.1 | 0.9 | 40.5 | 39.5 | 41.3 | 40.9 | 40.7 | 40.6 | 0.7 |
| | LogicVista | 56.8 | 56.6 | 60.0 | 58.2 | 59.1 | 58.1 | 1.4 | 61.3 | 60.2 | 57.7 | 59.3 | 57.7 | 59.2 | 1.6 |
| | MathVerse$_{vision\_only}$ | 61.9 | 63.7 | 62.6 | 64.5 | 64.0 | 63.3 | 1.0 | 67.0 | 67.1 | 66.2 | 65.4 | 66.6 | 66.5 | 0.7 |
| | MathVision | 46.8 | 46.9 | 47.8 | 47.7 | 47.4 | 47.3 | 0.4 | 50.0 | 50.0 | 50.4 | 50.4 | 50.1 | 50.2 | 0.2 |
| | MathVista$_{mini}$ | 78.6 | 78.7 | 79.0 | 79.1 | 78.7 | 78.8 | 0.2 | 81.4 | 79.6 | 79.9 | 79.9 | 81.3 | 80.4 | 0.9 |
| | WeMath | 55.0 | 55.6 | 53.6 | 54.5 | 56.2 | 55.0 | 1.0 | 59.8 | 58.5 | 57.3 | 60.6 | 57.8 | 58.8 | 1.4 |
| Average | All Benchmarks | 68.6 | 68.5 | 68.6 | 68.5 | 68.7 | 68.6 | 0.1 | 70.2 | 70.2 | 70.0 | 70.2 | 70.0 | 70.1 | 0.1 |

factual consistency with the ground truth. Crucially, evaluators overwhelmingly preferred the enriched data in Reasoning (72.74%) and Expression Style (69.92%). These findings confirm that our pipeline successfully transforms raw data into high-utility, logically structured content that meets human expectations for advanced reasoning tasks.

Table 9: Blind human evaluation results comparing Original vs. Enriched responses (532 pairwise comparisons). The high Tie rate in Accuracy confirms fidelity, while the strong preference for Enriched in Reasoning and Style demonstrates that our data aligns better with human standards for high-quality responses.

| Metric | Accuracy | Reasoning & Explanation | Instruction Following | Expression Style |
|---|---|---|---|---|
| Prefer Enriched | 42 (7.89%) | **387 (72.74%)** | 163 (30.64%) | **372 (69.92%)** |
| Prefer Original | 45 (8.46%) | 75 (14.10%) | 45 (8.46%) | 87 (16.35%) |
| Tie | **445 (83.65%)** | 70 (13.16%) | 324 (60.90%) | 73 (13.72%) |

## A.5 DATA CONTAMINATION ANALYSIS

To rigorously assess potential data leakage, we performed a comprehensive decontamination analysis against all 66,682 samples in our evaluation sets. We utilized Perceptual Hashing (pHash) for visual content and SimHash for textual instructions, calculating Hamming distances to identify matches ranging from exact duplicates (Distance = 0) to high-similarity near-duplicates (Distance $\leq 3$).

As detailed in Tab. 10, the detected leakage is extremely minimal. Even under the relaxed threshold, only 29 samples ($< 0.05\%$) were identified as potential overlaps, with merely 2 exact matches. No-

tably, these overlaps are concentrated in specific benchmarks like MathVision, while our primary comprehensive benchmarks—including MMMU, MMMU-Pro, MMStar, and CharXiv—remain completely free of any overlap.

We conclude that the identified overlap of 29 samples across all evaluation sets is statistically negligible and insufficient to influence the performance metrics or the validity of our conclusions. We strictly clarify that evaluation data was never intentionally included in our training set. Given that our dataset aggregates open-source collections, these few overlapping samples likely originated from the web-crawled nature of the upstream data sources.

Table 10: Data decontamination analysis results. We report the number of overlapping samples between Honey-Data-15M and the evaluation sets (total 66,682 samples) at different Hamming Distance thresholds.

| Threshold | Overlaps | Ratio | Benchmarks with Overlap (Count) |
|---|---|---|---|
| Exact Match ($= 0$) | 2 | 0.003% | MathVision (1), MathVista (1) |
| Diff $\leq 1$ | 4 | 0.006% | MathVision (1), MathVista (1), DocVQA (1), InfoVQA (1) |
| Diff $\leq 2$ | 14 | 0.021% | MathVision (10), DocVQA (2), MathVista (1), InfoVQA (1) |
| Diff $\leq 3$ | 29 | 0.043% | MathVision (19), MathVista (4), DocVQA (3), ChartQA (1), InfoVQA (1), OCRBench (1) |

### A.6 INTER-MODEL AGREEMENT ON VERIFICATION

To validate the reliability of our data curation pipeline and address potential concerns regarding the bias of using a single verifier model, we conducted a comprehensive experiment evaluating inter-model agreement using a diverse set of verifier models. We sampled a random subset of approximately 83,800 raw samples from our pool and performed the Fidelity Verification process independently using three different models with varying scales and capabilities: the original verifier utilized in our pipeline (Qwen2.5-VL-72B), a smaller, efficiency-focused variant (Qwen2.5-VL-32B), and a significantly larger Mixture-of-Experts model (Qwen3-VL-235B-A22B-Instruct) serving as a strong reference capability upper bound.

The number of samples retained by each model and the intersection of samples retained by all three models are reported in Tab. 11. The results demonstrate a high degree of inter-model consistency. The intersection of 63,203 samples constitutes **96.5%** of the data retained by the powerful reference model (Qwen3-VL-235B-A22B-Instruct) and **91.3%** by our chosen verifier. This significant overlap across model scales (spanning from 32B to 235B) confirms that the identified inconsistencies are largely objective flaws rather than artifacts of specific model biases. These findings validate Qwen2.5-VL-72B as a robust and reliable verifier for our pipeline.

Table 11: Inter-model agreement analysis for Fidelity Verification across distinct model scales. The high intersection count demonstrates that our filtering criteria are robust and not heavily biased by a specific model.

| Metric | Qwen2.5-VL-72B (Ours) | Qwen2.5-VL-32B | Qwen3-VL-235B | Intersection (All 3) |
|---|---|---|---|---|
| **Retained Samples** | 69,198 | 65,054 | 65,468 | **63,203** |

## B THE USE OF LARGE LANGUAGE MODELS

In preparing this manuscript, we utilized Large Language Models (LLMs) as a general-purpose assistive tool. Specifically, we employed LLMs to refine and polish the language, improve clarity, and perform comprehensive grammar checks. The models were also used to ensure that our phrasing and word choices were idiomatic and aligned with standard scientific discourse. Additionally, we received assistance from the LLM for minor LaTeX formatting adjustments and engaged in discussions with it to brainstorm and select an appropriate name for our dataset and model.

## C    REPRODUCIBILITY STATEMENT

We are committed to the full reproducibility of our work and will publicly release our full-stack suite. This includes the Honey-Data-15M corpus, the final Bee-8B model weights, the intermediate checkpoints from our training recipe, the specific training data used for each intermediate stage, and the complete code for our HoneyPipe data pipeline and DataStudio framework.

## D    ETHICS STATEMENT

This work adheres to the ICLR Code of Ethics and involved no human or animal subjects. Our work relies on public academic datasets and MLLMs. We acknowledge that Honey-Data-15M may inherit biases from its public sources, despite our HoneyPipe pipeline's design to systematically filter noise, factual errors, and image-instruction mismatches. Users should be aware of these potential limitations. Similarly, Bee-8B may generate inaccurate or harmful content. We are publicly releasing our full suite (dataset, pipeline, model weights, etc.) to promote open academic research and strongly discourage malicious use. We are committed to transparency, releasing our full methodology to help the community understand, build upon, and mitigate the limitations of such models.

## E    LIMITATIONS

### E.1    LIMITATION OF FIDELITY VERIFICATION

Our data curation pipeline employs an automated "LLM-as-a-Judge" mechanism for Fidelity Verification. We explicitly acknowledge that this design necessitates a strategic trade-off between recall (the coverage of valid samples) and computational cost.

Specifically, relying on a single-pass model-based verifier to enforce consistency with the original ground truth functions as a strict filter. High-quality, logically correct CoT samples may be inadvertently discarded if they conflict with the original answer due to valid variations (e.g., formatting differences, unit conversions, or synonyms) that the verifier fails to align. While massive-scale human annotation or extensive rejection sampling (e.g., generating and verifying dozens of candidates per sample) could theoretically resolve this ambiguity and achieve near-perfect recall, such approaches are prohibitively expensive and computationally infeasible at the scale of tens of millions of samples.

Therefore, we made a conscious design choice to prioritize precision (ensuring the retained data is factually consistent) and cost-effectiveness over recall. This pragmatic approach sacrifices a fraction of valid enriched data but ensures that the pipeline remains scalable, reproducible, and accessible to the broader academic community without requiring industrial-scale resources.

Furthermore, regarding the potential bias of using a single verifier model, we acknowledge that an ensemble of diverse judges could theoretically offer higher robustness. However, for the specific task of checking factual consistency against a ground truth, strong MLLMs generally exhibit a high degree of inter-model agreement. The inconsistencies identified by our verifier typically reflect objective flaws or significant deviations rather than model-specific artifacts. Thus, employing a single, strong open-source model as a verifier represents an optimal balance point, minimizing engineering complexity while maintaining high data quality standards.

### E.2    LIMITATION OF EVALUATION PARADIGM

As MLLMs evolve from simple pattern matchers to models capable of complex CoT reasoning, traditional deterministic evaluation methods like Regex have become insufficient. Advanced models now generate verbose outputs that standard extraction scripts often fail to parse correctly, leading to false negatives even when the reasoning is sound. Consequently, adopting "LLM-as-a-Judge" to perform semantic consistency checking has become a necessary evolution for the field.

However, this shift introduces a new layer of uncertainty: evaluation becomes inherently non-deterministic and dependent on the chosen judge's capabilities. Unlike rigid code-based metrics, an LLM judge introduces subjectivity regarding formatting strictness and semantic equivalence, and may exhibit subtle biases such as self-preference. We acknowledge that performance metrics in the

era of reasoning models are no longer absolute but are bounded by the subjectivity of the evaluator. Establishing standardized, objective, and automated metrics for rapidly evolving reasoning models remains a critical open challenge for the community.

# F    DETAILED TRAINING STAGE CONFIGURATIONS

This section provides an expanded description of the five-stage training recipe for Bee-8B, as introduced in Sec. 3. Our methodology follows a progressive curriculum, starting with foundational vision-language alignment, advancing to complex instruction-based SFT, and concluding with targeted refinement and reinforcement learning. For the purpose of clarity and reproducibility, the following subsections detail the specific data sources, and key configurations for each of the five training stages.

## F.1    STAGE 1: MLP WARMUP

This initial stage is dedicated to bridging the visual and language modalities. For this, we train only the MLP projector on a curated dataset of approximately one million image-caption pairs, keeping the vision encoder and LLM backbone frozen. This training set is a carefully assembled collection, combining roughly 560K samples from LLaVA-OneVision (Li et al., 2025a) with a high-quality subset of around 440K samples derived from COYO (Byeon et al., 2022). To generate this high-quality subset, we enriched the original responses by recaptioning them with the powerful Qwen2.5-VL-72B (Bai et al., 2025), followed by rule-based filtering. This entire process efficiently maps visual features into the language model's token space without disturbing the powerful pretrained weights of the core components.

## F.2    STAGE 2: VISION-LANGUAGE ALIGNMENT

In the second stage, we unfreeze all model components to build foundational multimodal capabilities. The training data is a large-scale composite resource, mixing approximately 12.6 million vision-language pairs with 1.43 million text-only samples to teach visual understanding while preserving language skills. The vision-language component is a meticulously curated collection from three main sources: a high-quality filtered subset of LAION (Schuhmann et al., 2022) (~6.9M), a refined selection from COYO (Byeon et al., 2022) (~5.4M), and additional specialized data from the stage 1.5 of LLaVA-OneVision (Li et al., 2025a) (~300K). All vision-language data was constructed using the same recaptioning and filtering methodology detailed in Appx. F.1. Crucially, to preserve the LLM's intrinsic reasoning abilities and mitigate catastrophic forgetting, we integrate a blend of text-only data from the Nemotron dataset (Bercovich et al., 2025; Nathawani et al., 2025). This blend includes long CoT samples for deep, multi-step problem-solving, sourced from reasoning-focused subsets like Nemotron-STEM (~458K), Nemotron-Math (~229K), and a reasoning-intensive version of Nemotron-Chat (~367K). This is complemented by the short CoT Nemotron-Chat subset (~376K) to maintain broad conversational proficiency. This mixed-modality approach enables the model to learn robust visual-language correlations while ensuring its core reasoning engine remains intact for subsequent stages.

## F.3    STAGE 3: MULTIMODAL SFT

In this stage, we train the model on the entire Honey-Data-15M to develop its advanced instruction-following and reasoning capabilities with vision. A detailed breakdown of the dataset's composition, including its sources and dual-level CoT distribution, is provided in Fig. 3 and Tab. 12. Training for one full epoch ensures complete exposure to this diverse data distribution. This is critical for learning from the rarer but more complex long CoT samples and allowing the model to learn their intricate reasoning patterns.

## F.4    STAGE 4: EFFICIENT REFINEMENT SFT

This final stage of SFT is designed for dual purposes: to conduct targeted refinement of the model's capabilities and to provide an accessible, efficient training option for researchers with limited computational resources. To this end, we curated a high-quality 1M subset from our full 15M dataset

Table 12: A detailed breakdown of the datasets used in our collection, categorized by task.

| Task | Dataset |
|---|---|
| General | SVIT-mix-665K (Zhao et al., 2023a), ALLaVA (Chen et al., 2024a), TQA (Kembhavi et al., 2017), LLaVA-NeXT-Data (Liu et al., 2024c), IconQA (Lu et al., 2021b), Objects365 (Shao et al., 2019), Vision FLAN (Xu et al., 2024), idefics375k (Laurençon et al., 2023), ViQuAE (Lerner et al., 2022), Co-Instruct (Wu et al., 2024), LVIS-InstructV4 (Wang et al., 2023a), DreamSim (Fu et al., 2023), SVIT-core-150K (Zhao et al., 2023a), Visual7W (Zhu et al., 2016), ShareGPT4V(SAM) (Li et al., 2025a), Cambrian (Filter) (Tong et al., 2024a), PixMo-CapQA (Deitke et al., 2025), HQ-Edit (Hui et al., 2025), PixMo-AskModelAnything (Deitke et al., 2025), ShareGPT4o (Cui et al., 2024), IDK (Cha et al., 2024), EST-VQA (Wang et al., 2020), COCO (Lin et al., 2014), PixMo-Point-Explanations (Deitke et al., 2025), Birds-to-Words (Forbes et al., 2019), GQA (Hudson & Manning, 2019), NextQA (Xiao et al., 2021), ALFWorld (Shridhar et al., 2021), Cauldron(mulberry) (Laurençon et al., 2024b), VSR (Liu et al., 2023a). A-OKVQA (Schwenk et al., 2022), KVQA (Shah et al., 2019), ContrastiveCaption (Jiang et al., 2024), WebQA (Chang et al., 2022b), WildVision (Lu et al., 2024b), New Yorker Caption (Hessel et al., 2023), InternVL-SA-1B-Caption (Chen et al., 2023; 2024e), ShareGPT4V(Knowledge) (Chen et al., 2024b), LLaVA-Instruct-300k (Liu et al., 2023c), VIST (Huang et al., 2016), LRV Normal (Liu et al., 2023b), KonIQ-10k (Hosu et al., 2020), Hateful Memes (Kiela et al., 2020), ART500K (Mao et al., 2017), NLVR2 (Suhr et al., 2019), ScanQA (Azuma et al., 2022), MMChat-Twitter-Post (Li et al., 2024b), nuScenes (Caesar et al., 2020), FlintstonesSV (Gupta et al., 2018), MagicBrush (Zhang et al., 2023a) |
| Chart | TinyChart (Zhang et al., 2024a), DVQA (Kafle et al., 2018), UniChart (Masry et al., 2023), CoSyn(chart, table, diagram, graphic) (Yang et al., 2025c), ArxivQA (Li et al., 2024c), FigureQA (Kahou et al., 2018), MMTab (Zheng et al., 2024), PlotQA (Methani et al., 2020), UReader QA (Ye et al., 2023), RobuT WikiSQL (Zhao et al., 2023b), TabMWP (Lu et al., 2023), RobuT WTQ (Zhao et al., 2023b), UReader KG (Ye et al., 2023), Chart2Text (Obeid & Hoque, 2020), ChartQA (Masry et al., 2022), MMC-Instruction (Liu et al., 2024a), RobuT SQA (Zhao et al., 2023b), MAVIS-Function (Zhang et al., 2025b), VisText (Tang et al., 2023), MultiHiertt (Zhao et al., 2022), SciTSR (Chi et al., 2019), LRV Chart (Liu et al., 2023b), SimChart9K (Xia et al., 2023), Infographic (Mathew et al., 2022), HiTab (Cheng et al., 2022) |
| Caption | PixMo-Cap (Deitke et al., 2025), WIT (Srinivasan et al., 2021), ST-VQA (Biten et al., 2019b), COYO-Recaption (Byeon et al., 2022), Sherlock (Hessel et al., 2022) |
| STEM | VisualWebInstruct(filtered) (Li et al., 2025a), MapQA (Chang et al., 2022a), VizWiz (Gurari et al., 2018), MetaMathQA (Yu et al., 2024a), Geo170K (Gao et al., 2025), VisualWebInstruct (Jia et al., 2025), MathV360K(TQA) (Shi et al., 2024), AI2D (Kembhavi et al., 2016), GeomVerse (Kazemi et al., 2023), GeoQA+ (Chen et al., 2021), MAVIS-Geo (Zhang et al., 2025b), CMM-Math (Liu et al., 2024d), CoSyn(math, music, chemical, circuit) (Yang et al., 2025c), MAVIS-Metagen (Zhang et al., 2025b), PMC-VQA (Zhang et al., 2023b), PathVQA (He et al., 2020), InterGPS (Lu et al., 2021a), VQA-RAD (Lau et al., 2018), RAVEN (Zhang et al., 2019a), AI2D(GPT4V) (Kembhavi et al., 2016), Geometry3K (Lu et al., 2021a), AI2D(InternVL) (Kembhavi et al., 2016), MMChem (Li et al., 2025b), WebSight (Laurençon et al., 2024c), UniGeo (Chen et al., 2022), ScienceQA (Lu et al., 2022) |
| Document | Ureader Chart (Ye et al., 2023), OCR-VQA (Mishra et al., 2019), InfographicVQA (Mathew et al., 2022), CoSyn(document, nutrition) (Yang et al., 2025c), POIE (Kuang et al., 2023), EATEN (Guo et al., 2019), FinTabNet (Zheng et al., 2021), UreaderOCR (Ye et al., 2023), InfoVQA (Mathew et al., 2022), Docmatix (Laurençon et al., 2024a), DocVQA (Mathew et al., 2021), ScreenQA (Hsiao et al., 2025), TextVQA (Singh et al., 2019), DocReason (Hu et al., 2024), VisualMRC (Tanaka et al., 2021), LLaVAR GPT4 (Zhang et al., 2023c) |
| Grounding & Counting | CLEVR (Johnson et al., 2017), TallyQA (Acharya et al., 2019), VisualGenome (Krishna et al., 2017), TQA (Kembhavi et al., 2017), MovieNet (Huang et al., 2020), MathV360K(VQA-AS) (Shi et al., 2024), CLEVR-Math (Lindström & Abraham, 2022), Super-CLEVR (Li et al., 2023b), IconQA (Lu et al., 2021b), CLEVR-Change (Park et al., 2019) |
| OCR | K12 Printing (Li et al., 2025a), ArXiv OCR (Zanichelli, 2024), HME (Yuan et al., 2022), VCR-Wiki (Zhang et al., 2024c), Sroie (Huang et al., 2021), IIIT 5K (Mishra et al., 2012), ICDAR-LSVT-zh (Sun et al., 2019), TextOCR (Singh et al., 2021), ReCTs (Zhang et al., 2019b), Orand-Car-A (Russakovsky et al., 2015), Rendered Text (Wendler, 2023), ICDAR2017 (Shi et al., 2017), Chrome-Writing (Wendler, 2023), MTWI(zh) (He et al., 2018), IAM (Marti & Bunke, 2002), ICDAR2019 (Biten et al., 2019a), CTW (Yuan et al., 2019) |

through a meticulous, multi-faceted selection strategy. Our process began by defining target proportions for various topics to ensure a balanced and comprehensive training corpus. We established a more rational distribution among topics such as STEM, Chart, Document, Grounding, and OCR, among others, while also maintaining a substantial portion for General. Furthermore, we aimed for

an approximate 1:1 ratio between long-chain and short-chain conversations to balance depth and breadth.

The core of our selection methodology was a quality-driven quota system. We first manually assigned a quality score (on a scale of 1 to 5) to each data source. This score was the primary factor used to determine each source's proportional contribution towards the overall target for its topic. This approach ensured that higher-quality sources contributed a proportionally larger number of samples, while still guaranteeing that every source was represented in the final subset to preserve diversity. Within each source's assigned quota, we employed a hybrid sampling strategy to balance difficulty and variety. Specifically, 60% of the data was selected by prioritizing conversations with the longest responses, based on the hypothesis that longer responses often correlate with more complex and challenging user queries. The remaining 40% was chosen via random sampling from the source to maintain broad diversity and prevent overfitting on specific types of difficult instructions. Through this stratified approach, we successfully constructed a 1M subset that is not only computationally efficient for training but also features a more rational topic distribution, preserving the difficulty and diversity of the original, larger dataset.

### F.5 STAGE 5: REINFORCEMENT LEARNING WITH GRPO

In this final stage, we employ the Group Relative Policy Optimization (GRPO) (Shao et al., 2024) algorithm to address persistent SFT issues such as text repetition, incomplete responses, and improper formatting. The reinforcement learning (RL) process is implemented with the verl framework (Sheng et al., 2025), where each rollout uses a batch size of 512, and the policy model updates its gradients with a batch size of 128.

The training data was constructed using prompts from the open-source MMK12 (Meng et al., 2025) and ViRL39K (Wang et al., 2025a) datasets. To improve data quality, the ViRL39K dataset was preprocessed by removing multi-image samples and randomly splitting 95% of the remaining data as the training set. For each prompt, we generated a set of candidate responses, enabling GRPO to refine the model's policy by learning to distinguish high-quality responses from flawed ones.

To guide this optimization, we adopt a rule-based reward function consisting of two components: a format reward (with a weight of 0.2) that enforces the presence of \boxed{} in the final output, and an accuracy reward (with a weight of 0.8) that evaluates whether the extracted content inside \boxed{} matches the ground-truth answer. This targeted reinforcement learning step provides a final polish, significantly improving the model's overall reliability and output quality.

## G   DATA LICENSING AND GOVERNANCE

To ensure rigorous compliance and promote responsible adoption within the community, we have established a transparent licensing framework and governance policy for Honey-Data-15M. Our approach is modeled after industry standards, such as The Cauldron Dataset (Laurençon et al., 2024b), to clarify rights and usage boundaries. Honey-Data-15M is primarily aggregated from established, large-scale open-source collections, including LLaVA-OneVision and MAmmoth-VL, which typically operate under permissive licenses like Apache 2.0. While we accessed data through these aggregates, we performed fine-grained tracing to generate the source distribution statistics, ensuring maximum credit is given to the original content creators.

We implement a clear dual-layered licensing structure for the repository. First, we explicitly state that Honey-Data-15M is a composite collection where each sub-dataset remains governed by its specific original license. Users are strictly mandated to adhere to the terms and restrictions of each original source. Second, to the extent that we hold rights in the modified prompts, structural formatting, and the newly generated CoT responses, these contributions are licensed under **CC-BY-NC-4.0** (Creative Commons Attribution-NonCommercial 4.0 International).

Regarding specific materials such as OCRed textbooks or K-12 content, we clarify that we did not independently digitize physical materials; we utilized them strictly as they exist in public academic datasets. For Personally Identifiable Information (PII), we primarily rely on the anonymization measures performed by the original dataset curators (e.g., SROIE explicitly addresses privacy). To address any potential future concerns regarding copyright or privacy, we have implemented a strict

**Notice and Takedown Policy**. We have established a dedicated communication channel for these concerns, and upon receiving a valid notice regarding infringement or PII exposure, we are committed to promptly reviewing the request and removing the contested content from our distribution.

From Tab. 13 to Tab. 19 provides a detailed enumeration of the license terms, usage restrictions (e.g., Non-Commercial) for the constituent source categories. For instances where definitive licensing information remained unverifiable despite exhaustive search, entries are left blank to maintain accuracy.

Table 13: Detailed overview of the licensing terms, usage restrictions for the General Category

| Dataset | License | Restrictions |
|---|---|---|
| SVIT-mix-665K | CC BY 4.0 | Appropriate credit must be given when sharing or adapting the work. |
| SVIT-core-150K | CC BY 4.0 | Appropriate credit must be given when sharing or adapting the work. |
| ALLaVA | CC BY-NC 4.0 | Appropriate credit must be given and no commercial use is permitted when sharing or adapting the work. |
| LLaVA-NeXT-Data | Apache-2.0 | Appropriate credit must be given and any modified versions must be licensed under the same terms when sharing or adapting the work. |
| LLaVA-Instruct-300k | CC BY 4.0 | Appropriate credit must be given when sharing or adapting the work. |
| Vision FLAN | – | – |
| idefics375k | CC BY 4.0 | Appropriate credit must be given when sharing or adapting the work. |
| PixMo-CapQ | ODC-By 1.0 | Appropriate credit must be given when using, sharing, or creating works from the database. |
| PixMo-AskModelAnything | ODC-By 1.0 | Appropriate credit must be given when using, sharing, or creating works from the database. |
| PixMo-Point-Explanations | ODC-By 1.0 | Appropriate credit must be given when using, sharing, or creating works from the database. |
| LVIS-Instruct4V | – | – |
| ShareGPT4V(SAM) | Apache-2.0 | Appropriate credit must be given and any modified versions must be licensed under the same terms when sharing or adapting the work. |
| Cambrian (Filter) | Apache-2.0 | Appropriate credit must be given and any modified versions must be licensed under the same terms when sharing or adapting the work. |
| ShareGPT4o | MIT | Appropriate credit must be given and the copyright and permission notice must be included when distributing the work. |
| NLVR2 | – | – |
| GQA | – | – |
| LRV Normal | BSD-3-Clause | Appropriate credit must be given, notices must be retained, and the author's name cannot be used for endorsement. |
| HQ-Edit | CC BY-NC 4.0 | Appropriate credit must be given and no commercial use is permitted when sharing or adapting the work. |
| ALFWorld | MIT | Appropriate credit must be given and the copyright and permission notice must be included when distributing the work. |
| Visual7W | MIT | Appropriate credit must be given and the copyright and permission notice must be included when distributing the work. |
| Co-Instruct | MIT | Appropriate credit must be given and the copyright and permission notice must be included when distributing the work. |
| Cauldron(mulberry) | – | – |
| A-OKVQA | Apache-2.0 | Appropriate credit must be given and any modified versions must be licensed under the same terms when sharing or adapting the work. |
| | | Continued on next page |

Table 13 – continued from previous page

| Dataset | License | Restrictions |
|---|---|---|
| IconQA | CC BY-NC-SA 4.0 | Appropriate credit must be given, no commercial use is permitted, and adaptations must be shared under the same license. |
| VIST | – | – |
| KVQA | – | – |
| ContrastiveCaption | Apache-2.0 | Appropriate credit must be given and any modified versions must be licensed under the same terms when sharing or adapting the work. |
| FlintstonesSV | – | – |
| InternVL-SA-1B-Caption | MIT | Appropriate credit must be given and the copyright and permission notice must be included when distributing the work. |
| IDK | BSD-3-Clause | Appropriate credit must be given, notices must be retained, and the author's name cannot be used for endorsement. |
| COCO | CC BY 4.0 | Appropriate credit must be given when sharing or adapting the work. |
| EST-VQA | – | – |
| Birds-to-Words | CC BY-SA 4.0 | Appropriate credit must be given and adaptations must be shared under the same license. |
| ART500K | – | Use is limited to non-commercial research, and no redistribution or commercial exploitation is permitted. |
| DreamSim | MIT | Appropriate credit must be given and the copyright and permission notice must be included when distributing the work. |
| ScanQA | CC BY-NC-SA 3.0 | Appropriate credit must be given, no commercial use is permitted, and adaptations must be shared under the same license. |
| MagicBrush | CC BY 4.0 | Appropriate credit must be given when sharing or adapting the work. |
| KonIQ-10k | MIT | Appropriate credit must be given and the copyright and permission notice must be included when distributing the work. |
| Hateful Memes | – | – |
| WebQA | CC0-1.0 | No restrictions are imposed, and the work may be used freely for any purpose. |
| nuScenes | CC BY-NC-SA 4.0 | Appropriate credit must be given, no commercial use is permitted, and adaptations must be shared under the same license. |
| Objects365 | CC BY 4.0 | Appropriate credit must be given when sharing or adapting the work. |
| MMChat-Twitter-Post | CC BY 4.0 | Appropriate credit must be given when sharing or adapting the work. |
| NextQA | MIT | Appropriate credit must be given and the copyright and permission notice must be included when distributing the work. |
| VSR | CC BY 4.0 | Appropriate credit must be given when sharing or adapting the work. |
| New Yorker Caption | CC BY 4.0 | Appropriate credit must be given when sharing or adapting the work. |
| ViQuAE | CC BY 4.0 | Appropriate credit must be given when sharing or adapting the work. |
| TQA | CC BY-NC 3.0 | Appropriate credit must be given and no commercial use is permitted when sharing or adapting the work. |
| ShareGPT4V(Knowledge) | CC BY-NC 4.0 | Appropriate credit must be given and no commercial use is permitted when sharing or adapting the work. |
| WildVision | CC BY 4.0 | Appropriate credit must be given when sharing or adapting the work. |

Table 14: Detailed overview of the licensing terms, usage restrictions for the Chart Category

| Dataset | License | Restrictions |
|---|---|---|
| TinyChart | Apache-2.0 | Appropriate credit must be given and any modified versions must be licensed under the same terms when sharing or adapting the work. |
| DVQA | CC BY-NC 4.0 | Appropriate credit must be given and no commercial use is permitted when sharing or adapting the work. |
| UniChart | – | – |
| CoSyn(chart) | ODC-By 1.0 | Appropriate credit must be given when using, sharing, or creating works from the database. |
| CoSyn(table) | ODC-By 1.0 | Appropriate credit must be given when using, sharing, or creating works from the database. |
| CoSyn(diagram) | ODC-By 1.0 | Appropriate credit must be given when using, sharing, or creating works from the database. |
| CoSyn(graphic) | ODC-By 1.0 | Appropriate credit must be given when using, sharing, or creating works from the database. |
| ArxivQA | CC BY-SA 4.0 | Appropriate credit must be given and adaptations must be shared under the same license. |
| FigureQA | MIT | Appropriate credit must be given and the copyright and permission notice must be included when distributing the work. |
| MMTab | MIT | Appropriate credit must be given and the copyright and permission notice must be included when distributing the work. |
| PlotQA | MIT | Appropriate credit must be given and the copyright and permission notice must be included when distributing the work. |
| UReader QA | Apache-2.0 | Appropriate credit must be given and any modified versions must be licensed under the same terms when sharing or adapting the work. |
| UReader KG | – | – |
| RobuT WikiSQL | MIT | Appropriate credit must be given and the copyright and permission notice must be included when distributing the work. |
| RobuT SQA | MIT | Appropriate credit must be given and the copyright and permission notice must be included when distributing the work. |
| RobuT WTQ | MIT | Appropriate credit must be given and the copyright and permission notice must be included when distributing the work. |
| TabMWP | CC BY-NC-SA 4.0 | Appropriate credit must be given, no commercial use is permitted, and adaptations must be shared under the same license. |
| Chart2Text | – | – |
| ChartQA | GPL-3.0 | Appropriate credit must be given and any modified versions must be released under the same license when distributing the work. |
| MMC-Instruction | CC BY-SA 4.0 | Appropriate credit must be given and adaptations must be shared under the same license. |
| MAVIS-Function | MIT | Appropriate credit must be given and the copyright and permission notice must be included when distributing the work. |
| VisText | GPL-3.0 | Appropriate credit must be given and any modified versions must be released under the same license when distributing the work. |
| SciTSR | MIT | Appropriate credit must be given and the copyright and permission notice must be included when distributing the work. |
| MultiHiertt | MIT | Appropriate credit must be given and the copyright and permission notice must be included when distributing the work. |
| | | Continued on next page |

Table 14 – continued from previous page

| Dataset | License | Restrictions |
|---|---|---|
| SimChart9K | – | – |
| HiTab | C-UDA 1.0 | Use is limited to computational purposes, attribution must be retained, and any redistributed data must remain under the same terms. |
| LRV Chart | BSD-3-Clause | Appropriate credit must be given, notices must be retained, and the author's name cannot be used for endorsement. |
| Infographic | – | – |

Table 15: Detailed overview of the licensing terms, usage restrictions for the Caption Category

| Dataset | License | Restrictions |
|---|---|---|
| COYO-Recaption | CC BY 4.0 | Appropriate credit must be given when sharing or adapting the work. |
| PixMo-Cap | ODC-By 1.0 | Appropriate credit must be given when using, sharing, or creating works from the database. |
| WIT | CC BY-SA 3.0 | Appropriate credit must be given and adaptations must be shared under the same license. |
| Sherlock | CC BY 4.0 | Appropriate credit must be given when sharing or adapting the work. |
| ST-VQA | – | – |

Table 16: Detailed overview of the licensing terms, usage restrictions for the STEM Category

| Dataset | License | Restrictions |
|---|---|---|
| VisualWebInstruct(filtered) | Apache-2.0 | Appropriate credit must be given and any modified versions must be licensed under the same terms when sharing or adapting the work. |
| MapQA | CC BY-NC-SA 4.0 | Appropriate credit must be given, no commercial use is permitted, and adaptations must be shared under the same license. |
| VisualWebInstruct | Apache-2.0 | Appropriate credit must be given and any modified versions must be licensed under the same terms when sharing or adapting the work. |
| MetaMathQA | MIT | Appropriate credit must be given and the copyright and permission notice must be included when distributing the work. |
| Geo170K | – | – |
| MAVIS-Metagen | MIT | Appropriate credit must be given and the copyright and permission notice must be included when distributing the work. |
| GeoQA+ | – | – |
| MAVIS-Geo | MIT | Appropriate credit must be given and the copyright and permission notice must be included when distributing the work. |
| CoSyn(math) | ODC-By 1.0 | Appropriate credit must be given when using, sharing, or creating works from the database. |
| CoSyn(music) | ODC-By 1.0 | Appropriate credit must be given when using, sharing, or creating works from the database. |
| CoSyn(chemical) | ODC-By 1.0 | Appropriate credit must be given when using, sharing, or creating works from the database. |
| CoSyn(circuit) | ODC-By 1.0 | Appropriate credit must be given when using, sharing, or creating works from the database. |
| AI2D | – | – |
| PMC-VQA | CC BY-SA | Appropriate credit must be given and adaptations must be shared under the same license. |

Continued on next page

Table 16 – continued from previous page

| Dataset | License | Restrictions |
|---|---|---|
| RAVEN | GPL-3.0 | Appropriate credit must be given and any modified versions must be released under the same license when distributing the work. |
| PathVQA | MIT | Appropriate credit must be given and the copyright and permission notice must be included when distributing the work. |
| MathV360K(TQA) | Apache-2.0 | Appropriate credit must be given and any modified versions must be licensed under the same terms when sharing or adapting the work. |
| ScienceQA | CC BY-SA 4.0 | Appropriate credit must be given and adaptations must be shared under the same license. |
| Geometry3K | MIT | Appropriate credit must be given and the copyright and permission notice must be included when distributing the work. |
| AI2D(InternVL) | – | – |
| MMChem | – | – |
| WebSight | CC BY 4.0 | Appropriate credit must be given when sharing or adapting the work. |
| UniGeo | – | – |
| GeomVerse | – | – |
| AI2D(GPT4V) | – | – |
| VizWiz | CC BY 4.0 | Appropriate credit must be given when sharing or adapting the work. |
| VQA-RAD | CC0-1.0 | No restrictions are imposed, and the work may be used freely for any purpose. |
| CMM-Math | BSD-3-Clause | Appropriate credit must be given, notices must be retained, and the author's name cannot be used for endorsement. |
| InterGPS | MIT | Appropriate credit must be given and the copyright and permission notice must be included when distributing the work. |

Table 17: Detailed overview of the licensing terms, usage restrictions for the Document Category

| Dataset | License | Restrictions |
|---|---|---|
| Ureader Chart | Apache-2.0 | Appropriate credit must be given and any modified versions must be licensed under the same terms when sharing or adapting the work. |
| OCR-VQA | – | – |
| CoSyn(document) | ODC-By 1.0 | Appropriate credit must be given when using, sharing, or creating works from the database. |
| CoSyn(nutrition) | ODC-By 1.0 | Appropriate credit must be given when using, sharing, or creating works from the database. |
| ScreenQA | CC BY 4.0 | Appropriate credit must be given when sharing or adapting the work. |
| FinTabNet | – | – |
| TextVQA | CC BY 4.0 | Appropriate credit must be given when sharing or adapting the work. |
| EATEN | – | – |
| DocVQA | – | – |
| LLaVAR GPT4 | CC-BY-SA 4.0 | – |
| Docmatix | MIT | Appropriate credit must be given and the copyright and permission notice must be included when distributing the work. |
| InfoVQA | – | – |
| UreaderOCR | Apache-2.0 | Appropriate credit must be given and any modified versions must be licensed under the same terms when sharing or adapting the work. |
| | | Continued on next page |

Table 17 – continued from previous page

| Dataset | License | Restrictions |
|---------|---------|--------------|
| DocReason | Apache-2.0 | Appropriate credit must be given and any modified versions must be licensed under the same terms when sharing or adapting the work. |
| InfographicVQA | – | – |
| VisualMRC | – | – |
| POIE | – | – |

Table 18: Detailed overview of the licensing terms, usage restrictions for the Grounding & Counting Category

| Dataset | License | Restrictions |
|---------|---------|--------------|
| CLEVR | CC BY 4.0 | Appropriate credit must be given when sharing or adapting the work. |
| TallyQA | Apache-2.0 | Appropriate credit must be given and any modified versions must be licensed under the same terms when sharing or adapting the work. |
| VisualGenome | CC BY 4.0 | Appropriate credit must be given when sharing or adapting the work. |
| IconQA | CC BY-NC-SA 4.0 | Appropriate credit must be given, no commercial use is permitted, and adaptations must be shared under the same license. |
| TQA | CC BY-NC 3.0 | Appropriate credit must be given and no commercial use is permitted when sharing or adapting the work. |
| MovieNet | – | – |
| CLEVR-Math | CC BY 4.0 | Appropriate credit must be given when sharing or adapting the work. |
| Super-CLEVR | BSD | Appropriate credit must be given and the copyright notice must be retained when redistributing the work. |
| MathV360K(VQA-AS) | Apache-2.0 | Appropriate credit must be given and any modified versions must be licensed under the same terms when sharing or adapting the work. |
| CLEVR-Change | – | – |

Table 19: Detailed overview of the licensing terms, usage restrictions for the OCR Category

| Dataset | License | Restrictions |
|---------|---------|--------------|
| K12 Printing | Apache-2.0 | Appropriate credit must be given and any modified versions must be licensed under the same terms when sharing or adapting the work. |
| ArXiv OCR | – | – |
| HME | – | – |
| VCR-Wiki | CC BY-SA 4.0 | Appropriate credit must be given and adaptations must be shared under the same license. |
| TextOCR | CC BY 4.0 | Appropriate credit must be given when sharing or adapting the work. |
| Sroie | MIT | Appropriate credit must be given and the copyright and permission notice must be included when distributing the work. |
| ICDAR-LSVT-zh | – | – |
| ReCTs | – | – |
| CTW | CC BY-NC-SA 4.0 | Appropriate credit must be given, no commercial use is permitted, and adaptations must be shared under the same license. |
| Rendered Text | – | – |
| ICDAR2017 | – | – |
| Chrome-Writing | – | – |
| MTWI(zh) | – | – |
| Continued on next page | | |

Table 19 – continued from previous page

| Dataset | License | Restrictions |
|---|---|---|
| IAM | MIT | Appropriate credit must be given and the copyright and permission notice must be included when distributing the work. |
| ICDAR2019 | MIT | Appropriate credit must be given and the copyright and permission notice must be included when distributing the work. |
| Orand-Car-A | CC BY-NC-ND 4.0 | Appropriate credit must be given, no commercial use is permitted, and no adaptations are allowed. |
| IIIT 5K | – | – |

# H    EVALUATION

We comprehensively evaluate our model across a wide range of benchmarks, which are grouped into three main categories: General VQA tasks, Table & Chart & OCR tasks, and Math & Reasoning tasks. The full list of benchmarks and their corresponding categories is presented in  Tab. 20.

## H.1    EVALUATION DETAIL

Our model is evaluated under two distinct inference configurations to assess its capabilities under different conditions. For the non-thinking mode, we employ a deterministic setting with the temperature set to 0 and a maximum output length of 8,192 tokens. Conversely, for the thinking mode, which is designed to elicit more detailed reasoning, we set the temperature to 0.6 and increase the maximum token length to 16,384.

All experiments are conducted using our customized version of the VLMEvalKit (Duan et al., 2024) framework. To ensure a more comprehensive and accurate evaluation, we introduced several key modifications. First, we extended the framework's LLM-based judging capabilities to several benchmarks that originally lacked this support, including ChartQA (Masry et al., 2022), In-foVQA (Mathew et al., 2022), DocVQA (Mathew et al., 2021), and CountBench (Paiss et al., 2023). Second, for the MathVerse (Zhang et al., 2024b) benchmark, we identified and addressed issues of inaccurate answer extraction and judging errors present in the original implementation. To enhance its robustness, we supplemented its judge with a broader set of test cases, thereby improving the reliability of the evaluation for this benchmark.

## H.2    PROMPT FOR LLM-BASED EVALUATION

Finally, to ensure transparency and reproducibility in our evaluation process, we provide the exact prompt from VLMEvalkit official (Duan et al., 2024) used for our LLM-based judging across the aforementioned benchmarks. This prompt constitutes a rigorous instruction set that directs the judge model to strictly evaluate the correctness of the candidate's final answer against the standard answer, ignoring intermediate reasoning errors and accommodating various valid output formats. The full prompt is detailed below:

---

**Prompt for LLM-based Answer Grading**

Please as a grading expert, judge whether the final answers given by the candidates below are consistent with the standard answers, that is, whether the candidates answered correctly. Here are some evaluation criteria:

1. Please refer to the given standard answer. You don't need to re-generate the answer to the question because the standard answer has been given. You only need to judge whether the candidate's answer is consistent with the standard answer according to the form of the question. **THE STANDARD ANSWER IS ALWAYS CORRECT AND THE QUESTION IS PERFECTLY VALID. NEVER QUESTION THEM.**

2. **ONLY** compare the **FINAL ANSWER** - **COMPLETELY IGNORE** any potential errors in the **REASONING PROCESSES**.

3. Some answers may be expressed in different ways, such as some answers may be a mathematical expression, some answers may be a textual description, as long as the meaning expressed is the same. Before making a judgment, please understand the question and the standard answer first, and then judge whether the candidate's answer is correct. If the standard answer does not specify a unit, but the candidate's answer includes a unit that is correct for the value given, consider it correct.

4. Some answers may consist of multiple items, such as multiple-choice questions, multiple-select questions, fill-in-the-blank questions, etc. Regardless of the question type, the final answer will be considered correct as long as it matches the standard answer, regardless of whether the reasoning process is correct. For multiple-select questions and multi-blank fill-in-the-blank questions, all corresponding options or blanks must be answered correctly and match the standard answer exactly to be deemed correct.

5. If the prediction is given with \boxed{}, please ignore the \boxed{} and only judge whether the candidate's answer is consistent with the standard answer.

6. If the candidate's answer is invalid (e.g., incomplete (cut off mid-response), lots of unnormal repetitive content, or irrelevant to the question, saying it can't answer the question because some irresistible factors, like ethical issues, no enough information, etc.), select option C (INVALID).

Please judge whether the following answers are consistent with the standard answer based on the above criteria. Grade the predicted answer of this new question as one of:
A: CORRECT
B: INCORRECT
C: INVALID

Just return the letters "A", "B", or "C", with no text around it. Here is your task. Simply reply with either CORRECT, INCORRECT, or INVALID. Don't apologize or correct yourself if there was a mistake; we are just trying to grade the answer.

<Original Question Begin>:
{question}
<Original Question End>
<Standard Answer Begin>:
{gold_answer}
<Standard Answer End>
<Candidate's Answer Begin>:
{llm_response}
<Candidate's Answer End>

Judging the correctness of the candidate's answer:

---

## H.3 BENCHMARKS

We comprehensively evaluate our model across a wide range of benchmarks, which are grouped into three main categories: General VQA tasks, Table & Chart & OCR tasks, and Math & Reasoning tasks. The full list of benchmarks and their corresponding categories is presented in Tab. 20.

Table 20: The evaluation benchmarks used in our study, grouped into three main categories: General VQA, Table & Chart & OCR, and Math & Reasoning.

| Task | Benchmark |
|---|---|
| General VQA | MMMU (Yue et al., 2024), AI2D (Kembhavi et al., 2016), MMStar (Chen et al., 2024c), MMVet (Yu et al., 2024b), HallusionBench (Guan et al., 2024), MMBench (Liu et al., 2024e), MMMU-Pro (Yue et al., 2025a), MMVP (Tong et al., 2024b), POPE (Li et al., 2023a) VisuLogic (Xu et al., 2025), RealWorldQA (xAI, 2024), CountBench (Paiss et al., 2023), BLINK (Fu et al., 2024), MMT-Bench (Ying et al., 2024), VLMs are Blind (Rahmanzadehgervi et al., 2024) |
| Table & Chart & OCR | DocVQA (Mathew et al., 2021), InfoVQA (Mathew et al., 2022), CharXiv (Wang et al., 2024c), SEED-Bench2-Plus (Li et al., 2024a), ChartQA (Masry et al., 2022), OCRBench (Liu et al., 2024f) |
| Math & Reasoning | MathVista (Lu et al., 2024a), MathVision (Wang et al., 2024a), MathVerse (Zhang et al., 2024b), LogicVista (Xiao et al., 2024), WeMath (Qiao et al., 2025), DynaMath (Zou et al., 2025) |

# I   PROMPT

In this section, we present the complete prompts used in our HoneyPipe. Specifically, two distinct prompts are utilized: one is for the Noise and Irrelevance Filtering process in Sec. 2.2, and the other is for the Fidelity Verification process in Sec. 2.3.

**Prompt for Noise and Irrelevance Filtering**

Given the image and the [QUESTION x], your task is to determine:
1. Whether the image and question are relevant to each other;
2. Whether they form an appropriate question;
3. Whether there are any obvious issues with the question.

Below are some examples of evaluation scenarios:
1. **Relevance Check**: Is the image related to the question?
- Example 1: If the question is about "the dog in the photo," the image should clearly depict a dog or something directly related to dogs.
- Example 2: If the question asks about the diameter of a circle in the image, but the image does not contain a circle → filter out.
- Example 3: If the image is completely unrelated to the question → filter out.
2. **Ambiguous or Vague Question**: If the question is too vague, unclear, or lacks sufficient context to establish a clear connection with the image → filter out.
- Example 1: "What do you think about this?" without specifying what "this" refers to in the image's context.
- Example 2: Ambiguous question (e.g., unclear referents) → filter out.
- Example 3: The question is not a question. (e.g., a list of numbers, a table, etc.) → filter out.
3. **Language Filter**: If the question contains languages other than Chinese or English → filter out.

After evaluating the image and question, provide a clear decision:
- false: If the image and question are relevant, the relationship is clear, and no issues are found.
- true: If the image and question do not match, contain issues (e.g., ambiguity, contradiction, irrelevance, language errors), or fail any evaluation criteria.

**Output Format(json format)**:
```json
{
    "q0": "true",
    "q0_reason": "Briefly explain the reason for filtering",
    ...
}
```

where the returned result index starts from 0. If there are N questions, return the results of all questions(q0, q1, ..., qN).
Here are the specific [QUESTION x] and image:

**Prompt for Fidelity Verification**

**Task:** Evaluate whether the new answer [ANSWER x] should be retained or filtered compared to the original answer [ORI_ANSWER x] for [QUESTION x] according to the following guidelines. Provide the evaluation results in JSON format (where x is an index starting from 0).

**Evaluation Guidelines:**
1. **Open-Ended/Descriptive Questions**
- **Rule:** Answers are inherently diverse and subjective. Retain the new answer unless it is clearly irrelevant to the question.
2. **Precise Answer Questions**
- **Rule:** Retain the new answer if it is numerically equivalent to the original answer (e.g., different formats, minor rounding errors within 0.1, valid unit conversions). Filter it if there is a fundamental conflict in numerical values.
3. **Factual Questions**
- **Rule:** Retain the new answer if it conveys the same fact (rephrasing is allowed). Filter it if it introduces fabricated or irrelevant content.
4. **Conceptual or Common Sense Questions**
- **Rule:** Retain the new answer if the final conclusion or reasoning result is consistent, regardless of wording. Filter it if there is a logical contradiction or opposite judgment.
5. **Chart Analysis Questions**
- **Rule:** Retain the new answer if both reference consistent content in the chart. Filter it if it conflicts with the factual data in the chart.
6. **Logical Reasoning or Hypothetical Questions**
- **Rule:** Retain the new answer if the final conclusion is consistent, even if the steps differ. Filter it if the results are contradictory or contain logical errors.
7. **Sorting/Priority/Comparison Questions**
- **Rule:** Retain the new answer if the relative order or judgment is consistent. Filter it if the new answer reverses or distorts the sorting/comparison result.

**Judgment Criteria:**
- **false:** The new answer meets consistency requirements, supplements information, and should be retained.
- **true:** The new answer is inconsistent, irrelevant, or contains factual errors, and should be filtered.

**Output Format (JSON):**
```json
{
    "q0": "false",
    "q0_reason": "The new answer effectively rephrases the same factual information.",
    ...
    "qN": "true",
    "qN_reason": "The new answer introduces numerical values conflicting with the original precise answer."
}
```

**Explanation:**
- "false" indicates that the new answer should be retained (consistent), and "true" indicates that the new answer should be filtered (inconsistent or erroneous).
- The index starts from 0 and must cover all questions (q0, q1, . . . , qN).
- ...

Here are the specific [QUESTION x], [ANSWER x] and [ORI_ANSWER x]:

# J  NOISE DATA CASE STUDY

## J.1  FORMAT FLAW

Some data exhibit format flaws, such as missing instructions or responses. Below is an example of data filtered out due to a missing response.

---

**Case: Format flaw**

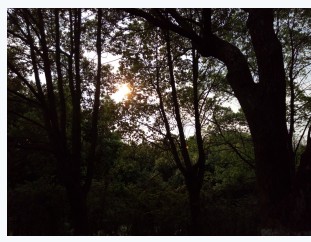

**User Question:**

What might be the implications of this lighting for a photographer?

**Response:**

(None)

**Filtering Reason:**

Only half conversation.

---

## J.2  LOW QUALITY IMAGE

Low-quality images, such as those with excessively low resolution, overly large aspect ratios, or blurriness, are detrimental to the model's ability to extract high-quality visual features and semantic information, thereby impairing the model's performance and effectiveness. Below is an example of an image filtered out due to excessively low resolution.

## J.3  TEXT REPETITION

Some responses generated by MLLMs may suffer from the issue of text repetition. Below is an example of data filtered out due to the inclusion of repeatedly occurring segments in the response. The first occurrence of a repeatedly appearing pattern is marked in red, and the second occurrence is marked in blue. More repeated occurrences have been omitted.

**Case: Low-quality image**

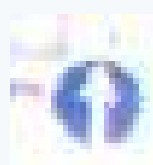

**User Question:**

In this task, you will classify traffic signs in the given image. The image is in color and contains various traffic signs captured in real-world conditions. Your goal is to accurately identify the type of traffic sign present in the image. Consider factors such as the sign's shape, color, and symbols to ensure correct classification. A sample output should look like this: 'Speed limit (80km/h)' Options: (a) Go straight or right (b) Right-of-way at the next intersection (c) Dangerous curve to the left (d) Bicycles crossing (e) Speed limit (30km/h) (f) Speed limit (120km/h) (g) Speed limit (50km/h) (h) End of speed limit (80km/h) (i) Pedestrians (j) Ahead only

**Response:**

(j) Ahead only

**Filtering Reason:**

Image size too small: [26, 27]

**Case: Text repetition**

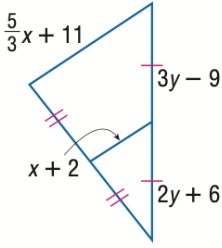

**User Question:**

Hint: Please answer the question and provide the correct option letter, e.g., A, B, C, D, at the end.
Question: Determine the value of x.
Options: (A) 12     (B) 15     (C) 18     (D) 21

**Response (part):**

To determine the value of $x$, we need to analyze the given geometric configuration. The image shows a triangle with segments marked with expressions involving $x$ and $y$.
...

$$x = -156$$

However, this result seems incorrect based on the provided options. Let's re-evaluate the steps. Revisiting the substitution step:

$$\frac{5}{3}x + 11 = \frac{3x - 12}{2} - 9$$

Multiply every term by 6:

$$10x + 66 = 9x - 36 - 54$$

$$\cdots$$

$$x = -156$$

This result is still incorrect. Let's re-evaluate the initial setup and equations.
Revisiting the initial equations:

$$\frac{5}{3}x + 11 = 3y - 9$$

$$\cdots$$

$$\frac{5}{3}x + 11 = \frac{3x - 12}{2} - 9$$

Multiply every term by 6:

$$10x + 66 = 9x - 36 - 54$$

$$\cdots$$

$$x = -156$$

This result is still incorrect. Let's re-evaluate the initial setup and equations.
Revisiting the initial equations:

$$\frac{5}{3}x + 11 = 3y - 9$$

...

## J.4 IMAGE-INSTRUCTION MISMATCH

Some data contain instructions irrelevant to the associated images, which is detrimental to the MLLM's ability to align visual features with semantic information. Below is an example of data filtered out due to the irrelevance between the image and its corresponding instruction.

---

**Case: Image-instruction mismatch**

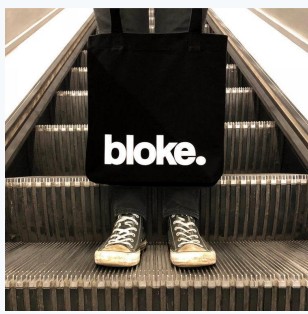

**User Question:**

What natural elements contribute to the sense of tranquility in this landscape, and how does the play of light enhance the overall atmosphere?

**Response:**

The natural elements that contribute to the tranquility include the lush green meadow, vibrant wildflowers, and the gently flowing stream. These elements evoke a sense of peace and harmony with nature. The play of light, with sunlight filtering through the tree branches, creates a dynamic interplay of shadows and highlights, further enhancing the serene atmosphere by adding depth and visual interest to the landscape.

**Filtering Reason:**

The image shows a person on an escalator with a bag labeled 'bloke,' which is unrelated to natural elements or tranquility in a landscape. The question and image are not relevant to each other.

---

## J.5 UNANSWERABLE INSTRUCTION

Some data contain overly vague instruction descriptions, or the image-instruction pairs lack sufficient information, rendering the instructions unanswerable. However, it is important to note that well-articulated open-ended instructions will still be retained. Below is an example of an image-instruction pair that was filtered out due to insufficient information caused by partial corruption of the image.

## J.6 LOW QUALITY INSTRUCTION

Some data contain images that are relevant to their corresponding instructions, but the instructions can be answered without extracting information from the images. Such data only leverages the LLM component of MLLMs and provides no benefit to the training of the vision encoder and projector. Below is an example of data filtered out because the instruction can be answered without extracting information from the image.

**Case: Unanswerable instruction**

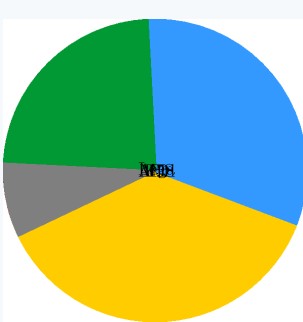

**User Question:**

Based on the pie chart representing the distribution of parliamentary seats in Italy, which party has the second-largest share, and what percentage of seats does it hold?

**Response:**

M5S, 25%

**Filtering Reason:**

The image is a pie chart, but it lacks labels or a legend to identify which colors correspond to which parties. Without this information, it is impossible to determine which party has the second-largest share or the percentage of seats it holds.

**Case: Low quality instruction**

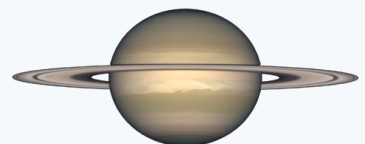

**User Question:**

Who discovered the moon of Saturn called Titan in 1655? Answer the question using a single word or phrase.

**Response:**

Huygens

**Filtering Reason:**

The image shows Saturn, but the question is about the discovery of Titan, which is not depicted in the image. The image does not provide any information relevant to answering the question.

## K QUALITATIVE ANALYSIS OF HONEY-DATA-15M

### K.1 FIDELITY VERIFICATION-FAILED DATA

The fidelity verification process in Sec. 2.3 filters out data where there are conflicts between the enriched response and the original response. Below is an example of a factual query being filtered out because the enriched response differs from the original response.

---

**Case: Factual inaccuracy**

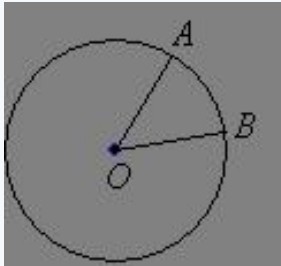

**User Question:**

Hint: Please answer the question and provide the final answer at the end. Question: What is the radius of the circle O if the length of arc AB is 4*pi and the measure of angle AOB is 40 degrees?

**Original Response:**

The answer is 9

**Rewritten Response:**

To find the radius of circle $O$, we can use the formula for the length of an arc. The formula for the length of an arc $s$ is:

$$s = \frac{\theta}{360°} \times 2\pi r$$

where $\theta$ is the central angle in degrees, and $r$ is the radius of the circle.
We know that the length of arc $AB$ is $4\pi$ and the central angle $\angle AOB$ is $40°$. Plugging these values into the formula:

$$4\pi = \frac{40°}{360°} \times 2\pi r$$

First, simplify $\frac{40°}{360°}$ to $\frac{1}{9}$:

$$4\pi = \frac{1}{9} \times 2\pi r$$

Simplify the right-hand side:

$$4\pi = \frac{2\pi r}{9}$$

To solve for $r$, multiply both sides of the equation by 9:

$$36\pi = 2\pi r$$

Then divide both sides by $2\pi$:

$$r = \frac{36\pi}{2\pi} = 18$$

Final answer: The radius of circle $O$ is 18.

**Filtering Reason:**

The new answer introduces a numerical value 18 that conflicts with the original precise answer 9.

---

## K.2  SHOWCASE OF HIGH-QUALITY ENRICHED DATA

In this section, we showcase data from diverse topics within our Honey-Data-15M. A subset of these responses with long Chain-of-Thought reasoning. The enriched responses incorporate more in-depth knowledge, feature more accurate responses, and present more detailed reasoning processes.

---

**Case: Caption (with short CoT)**

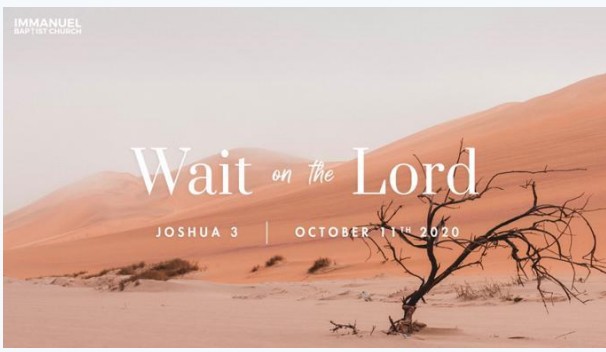

**User Question:**

Can you describe the scene depicted in the image?

**Original Response:**

The image shows a tree in the desert with a sand dune in the background.

**Enriched Response:**

The image depicts a serene and somewhat desolate desert scene with the following elements:

1. **Desert Landscape**:
   - The background features rolling sand dunes with a warm, orange hue, characteristic of a desert environment.
   - The sky is clear and light, suggesting a calm and open atmosphere.
2. **Tree**:
   - In the foreground, there is a bare, leafless tree with twisted branches. The tree appears to be struggling or enduring harsh conditions, symbolizing resilience or waiting.
3. **Text Overlay**:
   - The text "Wait on the Lord" is prominently displayed in the center of the image, written in a clean, white font.
   - Below the main text, there is additional information: "Joshua 3 | October 11th, 2020," indicating a reference to a biblical passage and a specific date, likely for a church service or event.
   - In the top left corner, the text "Immanuel Baptist Church" is visible, suggesting the image is related to a church event or sermon.

**Interpretation**:
The scene conveys a sense of patience, endurance, and faith, as the desert and the bare tree symbolize challenging conditions. The text "Wait on the Lord" reinforces the theme of waiting and trusting in divine guidance, aligning with the biblical reference to Joshua 3. The overall tone is reflective and spiritual, encouraging viewers to find strength and hope in difficult times.

---

**Case: General (with short CoT)**

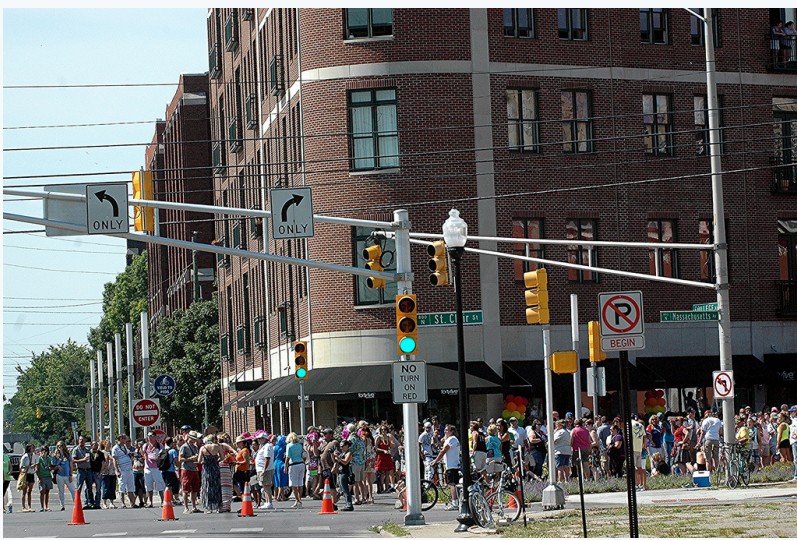

**User Question:**

Can you tell what the people are doing?

**Original Response:**

The people appear to be either walking or standing on the street, perhaps waiting to cross the intersection or attending a public event.

**Enriched Response:**

The image shows a large group of people gathered in a public space, likely on a street or near a building. Here's what can be inferred about their activities:

1. **Crowd Gathering**: The people are standing in a large group, suggesting they are attending an event or gathering. The density of the crowd indicates it might be a public or organized event.

2. **Event or Parade**: The presence of traffic cones, barricades, and the organized nature of the crowd suggest this could be a parade, festival, or public demonstration. The "BEGIN" sign on the pole might indicate the start of a route or event.

3. **Traffic Management**: The traffic lights and signs (e.g., "ONLY," "NO TURN ON RED") indicate that the area is being managed to control traffic flow, likely to accommodate the event.

4. **Casual Attire**: The people are dressed in casual clothing, which is typical for outdoor public events.

5. **Balloons and Decorations**: The balloons visible in the background suggest a festive or celebratory occasion.

Overall, the people appear to be participating in or observing a public event, such as a parade, festival, or community gathering. The setup indicates careful planning to manage the crowd and traffic.

**Case: Grounding & Counting (with short CoT)**

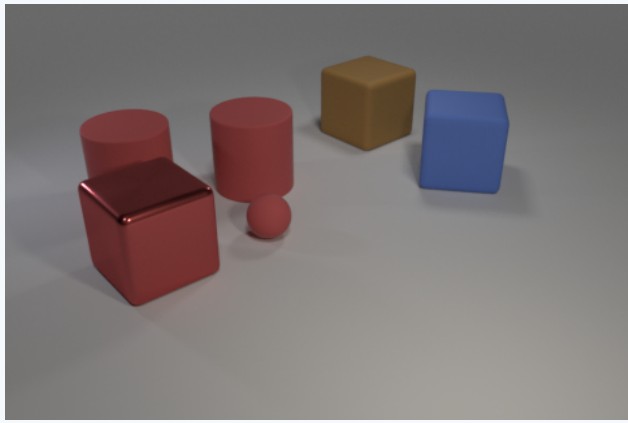

**User Question:**

Think out loud before giving the final conclusion. Is there a large blue cube made of the same material as the small sphere?

**Original Response:**

yes

**Enriched Response:**

To answer the question, let's analyze the image step by step:

1. **Identify the objects in the image**:
    - There are two red cylinders.
    - There is a red metallic cube.
    - There is a small red sphere.
    - There is a brown matte cube.
    - There is a large blue cube.

2. **Examine the material of the small sphere**:
    - The small sphere is red and appears to have a matte finish. This can be inferred from its lack of reflective properties and its uniform color.

3. **Examine the material of the large blue cube**:
    - The large blue cube also appears to have a matte finish. It does not reflect light in a way that suggests a metallic or glossy surface.

4. **Compare the materials**:
    - The small red sphere has a matte finish.
    - The large blue cube also has a matte finish.
    - Since both objects share the same matte appearance, it is reasonable to conclude that they are made of the same material.

5. **Final conclusion**:
    - There is a large blue cube, and it appears to be made of the same material as the small red sphere.

**Final Answer: Yes, there is a large blue cube made of the same material as the small sphere.**

**Case: Chart (with long CoT)**

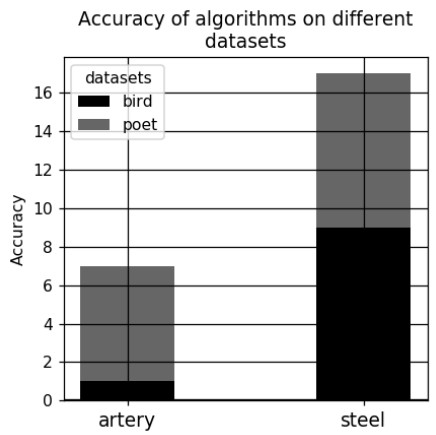

Accuracy of algorithms on different datasets

**User Question:**

Which algorithm has the smallest accuracy summed across all the datasets? Please conclude your answer as Answer: xxx at the end if possible.

**Enriched Response (Part):**

$<$ think $>$ So, let's solve this problem. First, we need to find the accuracy of each algorithm (artery and steel) across all datasets (bird and poet), sum them, and then compare. First, look at the "artery" algorithm. The black bar (bird dataset) has a height of 1, and the gray bar (poet dataset) has a height of 7 (since total for artery is 7, and bird is 1, so poet is 7-1=6... $<$ /think $>$

To determine which algorithm has the smallest accuracy summed across all datasets, we analyze the two algorithms (artery and steel) by calculating the total accuracy across both datasets (bird and poet) for each:

1. Artery algorithm:

   - Accuracy on the bird dataset: The height of the black bar is 1.
   - Accuracy on the poet dataset: The height of the gray bar is $7 - 1 = 6$ (since the total height of the artery bar is 7).
   - Sum of accuracies: $1 + 6 = 7$.

2. Steel algorithm:

   - Accuracy on the bird dataset: The height of the black bar is 9.
   - Accuracy on the poet dataset: The height of the gray bar is $17 - 9 = 8$ (since the total height of the steel bar is 17).
   - Sum of accuracies: $9 + 8 = 17$.

Comparing the two sums (7 for artery and 17 for steel), the smaller sum belongs to the artery algorithm.

Answer: artery

**Case: STEM (with long CoT)**

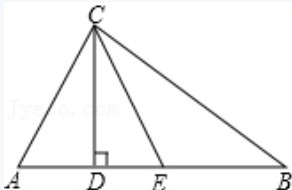

**User Question:**

In the right triangle ABC, ∠ACB = 90°, CD is the height from C to side AB, CE is the median to side AB, AD = 2, CE = 5, then CD = ( )
A. 2      B. 3      C. 4      D. $2\sqrt{3}$ .

**Enriched Response (Part):**

$< \text{think} >$ Got it, let's try to solve this problem step by step. First, let's recall some properties of right triangles, medians, and altitudes.
We have a right triangle $ABC$ with $\angle ACB = 90°$. $CE$ is the median to $AB$, so in a right triangle, the median to the hypotenuse is half the hypotenuse. That's a key property: in a right triangle, the median from the right angle to the hypotenuse is equal to half the hypotenuse. So $CE = \frac{1}{2}AB$. We know $CE = 5$, so $AB = 2 \times CE = 10$... $< /\text{think} >$
To solve the problem, we can use properties of right triangles, medians, and altitudes. Let's break down the steps:
Step 1: Use the property of the median in a right triangle
In a right triangle, the median to the hypotenuse is half the length of the hypotenuse.
Given $\triangle ABC$ is a right triangle with $\angle ACB = 90°$, and $CE$ is the median to $AB$, we have:

$$CE = \frac{1}{2}AB$$

Since $CE = 5$, substitute to find $AB$:

$$5 = \frac{1}{2}AB \implies AB = 10$$

Step 2: Analyze segments on $AB$
$E$ is the midpoint of $AB$ (because $CE$ is a median), so $AE = BE = \frac{AB}{2} = 5$.
We know $AD = 2$, so the segment $DE$ (between $D$ and $E$) is:

$$DE = AE - AD = 5 - 2 = 3$$

Step 3: Use the Pythagorean theorem in $\triangle CDE$
$CD$ is the altitude from $C$ to $AB$, so $CD \perp AB$, meaning $\angle CDE = 90°$. Thus, $\triangle CDE$ is a right triangle.
In $\triangle CDE$:

- Leg $DE = 3$.
- Hypotenuse $CE = 5$ (given)
- Leg $CD$ (unknown, let's call it $x$)

By the Pythagorean theorem:
$$CD^2 + DE^2 = CE^2$$

Substitute $DE = 3$ and $CE = 5$:

$$x^2 + 3^2 = 5^2 \implies x^2 + 9 = 25 \implies x^2 = 16 \implies x = 4$$

Thus, $CD = 4$, and the answer is $C$.

**Case: Document (with long CoT)**

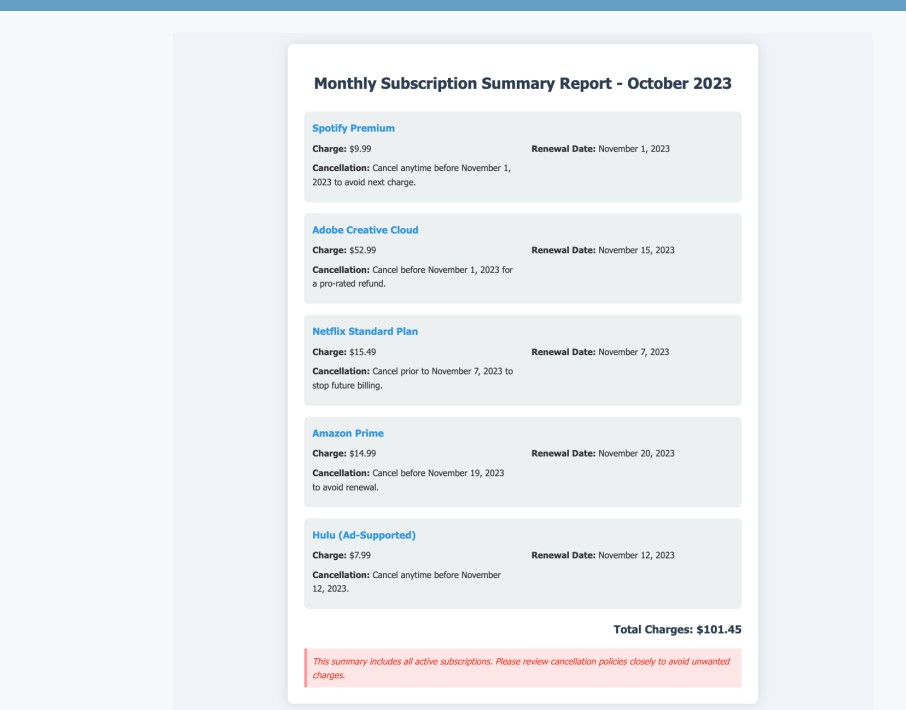

**User Question:**

Which service has the next renewal date after Spotify?

**Enriched Response:**

$<$ think $>$ So, let's look at the renewal dates for each service. First, Spotify's renewal date is November 1, 2023. Now we need to find the next one after that. Let's list all renewal dates:

- Spotify Premium: November 1, 2023
- Netflix Standard Plan: November 7, 2023
- Hulu (Ad-Supported): November 12, 2023
- Adobe Creative Cloud: November 15, 2023
- Amazon Prime: November 20, 2023

So after November 1, the next date is November 7, which is Netflix Standard Plan.
$<$ /think $>$
To determine which service has the next renewal date after Spotify, we first identify Spotify's renewal date and then compare it with the renewal dates of other services:

1. Locate Spotify Premium's renewal date: **November 1, 2023**.
2. List the renewal dates of all other services:
   - Netflix Standard Plan: November 7, 2023
   - Hulu (Ad - Supported): November 12, 2023
   - Adobe Creative Cloud: November 15, 2023
   - Amazon Prime: November 20, 2023

Identify the earliest date after November 1, 2023: November 7, 2023, which corresponds to **Netflix Standard Plan**.
Thus, the service with the next renewal date after Spotify is Netflix Standard Plan.

# L  QUALITATIVE SHOWCASE OF BEE-8B

**Case: STEM**

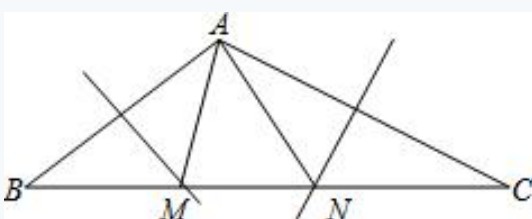

**User Question:**

Given: In △ABC, the perpendicular bisectors of AB and AC intersect BC at points M and N, respectively. AB = 4, AC = 7, BC = 10. Find the perimeter of △AMN.

**Bee-8B Response:**

$<$ think $>$ Got it, let's try to solve this problem step by step. First, let's recall what a perpendicular bisector is. A perpendicular bisector of a segment is a line that is perpendicular to the segment and divides it into two equal parts. So, if M is on the perpendicular bisector of AB, that means MA = MB. Similarly, N is on the perpendicular bisector of AC, so NA = NC.

Now, we need to find the perimeter of triangle AMN. The perimeter of a triangle is the sum of its sides, so that would be AM + MN + AN. Let's substitute the equal segments we just found. Since MA = MB and NA = NC, we can replace AM with MB and AN with NC. So the perimeter becomes MB + MN + NC... $<$ /think $>$

To solve the problem, we use the **property of perpendicular bisectors**: any point on the perpendicular bisector of a segment is equidistant from the segment's two endpoints.

**Step 1: Analyze the Perpendicular Bisectors**

- $M$ lies on the perpendicular bisector of $AB$, so $MA = MB$ (by the perpendicular bisector theorem).
- $N$ lies on the perpendicular bisector of $AC$, so $NA = NC$ (by the perpendicular bisector theorem).

**Step 2: Express the Perimeter of $\triangle AMN$**

The perimeter of $\triangle AMN$ is $AM + MN + AN$.

Substitute $AM = MB$ and $AN = NC$ (from Step 1):

$$\text{Perimeter of } \triangle AMN = MB + MN + NC$$

**Step 3: Relate to $BC$**

Points $M$ and $N$ lie on $BC$, so $MB + MN + NC$ is exactly the length of $BC$.
We know $BC = 10$ (given). Thus:

$$\text{Perimeter of } \triangle AMN = BC = 10$$

**Answer:** The perimeter of $\triangle AMN$ is **10**.

**Case: General**

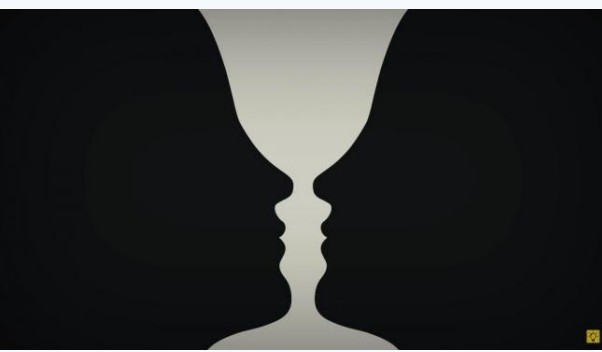

**User Question:**

This is an illusion image. What kind of visual illusions can it make people perceive? And why is it able to produce such effects?

**Bee-8B Response:**

< think >... < /think >
To analyze the visual illusions in this image and their underlying mechanisms, let's break it down:
**1. Types of Visual Illusions Perceived**
This image triggers **ambiguous shape perception** (a type of *multistable illusion*). Viewers can perceive the same visual contour in two distinct ways:

- **Interpretation 1: Two face profiles**

  The left and right sides of the gray shape resemble the side profiles of two human faces (from forehead to chin). The central "connection" between the two profiles creates the illusion of two faces facing each other.

- **Interpretation 2: A goblet (chalice) shape**

  The central curve and the extended top/bottom of the gray shape can also be read as the outline of a goblet: the central curve as the goblet's bowl, and the top/bottom as the rim and stem.

**2. Why It Produces These Effects**
The illusion arises from **cognitive biases** and **visual perception principles**, rooted in how the brain processes shape and familiarity:

- **Shape Similarity & Prior Experience**

  Human faces and goblets share *structural similarities* in their curved contours (e.g., the smooth, continuous curves of a face's profile match the curve of a goblet's bowl). The brain prioritizes interpreting ambiguous shapes using *prior experience*—we evolved to recognize faces (a "biologically significant" shape for survival) and are also trained to recognize everyday objects like goblets. When a single contour matches both sets of familiar shapes, the brain generates multiple plausible interpretations.

- **Gestalt Psychology: "Ambiguity" and "Continuity"**

  Gestalt principles explain how the brain organizes visual information into coherent wholes. Here, the *continuity* of the gray shape's curve is split into two competing "wholes": one resembling two faces (due to the brain's bias for face-like patterns) and one resembling a goblet (due to the brain's bias for object-like patterns). This split creates the illusion of simultaneous, conflicting interpretations.

In short, the image leverages **one contour's ability to fit two familiar shapes** (faces + goblet) and the brain's innate tendency to prioritize familiar, evolutionarily important shapes (like faces) to create a multistable, ambiguous perception.

**Case: Chart**

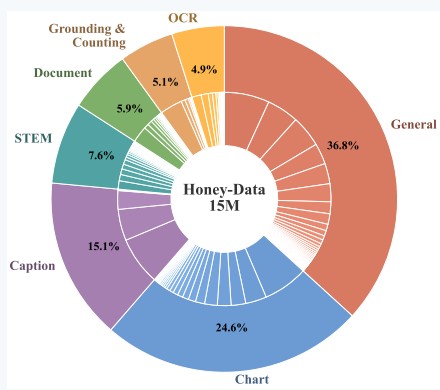

**User Question:**

If you want to build a new dataset based on this dataset, how would you adjust the distribution of the data?

**Bee-8B Response:**

< think > So, let's tackle this question. First, I need to understand the current distribution from the pie chart. The Honey-Data 15M dataset has segments: General (36.8%), Chart (24.6%), Caption (15.1%), STEM (7.6%), Document (5.9%), Grounding & Counting (5.1%), OCR (4.9%).... < /think >

To adjust the data distribution when building a new dataset based on *Honey-Data 15M*, follow this structured approach, which accounts for the **current distribution** (from the pie chart) and the **task's requirements**:

**1. Analyze the Current Distribution** ...

**2. Define the New Dataset's Purpose**

The distribution adjustment depends on the **task** the new dataset will support. For example:

- *Niche application*: If the task targets a rare category (e.g., *OCR* for document analysis), prioritize increasing that category's share.
- *Balanced general-purpose*: If the task requires broad coverage, rebalance dominant categories (*General, Chart*) and enhance smaller ones.
- *Multi-modal focus*: If the task combines categories (e.g., *Chart + Caption* for image-text models), balance those segments.

**3. Select Categories to Adjust**

Determine which categories need *oversampling* (more data) or *undersampling* (less data):

- **Oversample underserved categories**: If a category is critical but underrepresented (e.g., *OCR* at 4.9%), increase its proportion.
- **Undersample overrepresented categories**: If a dominant category (*General* at 36.8%) is less relevant to the task, reduce its share.

**4. Implement Adjustment Methods** ...

**5. Validate the Adjusted Distribution** ...

**Example: Adjusting for an OCR-Focused Task**

If the new dataset targets *OCR* (currently 4.9%):

1. **Oversample *OCR***: Collect more OCR samples (e.g., scanned documents, handwritten notes) and augment existing ones (e.g., add noise, change fonts).
2. **Undersample *General***: Reduce the *General* segment to free up space for *OCR* (e.g., scale *General* to 25% instead of 36.8%).
3. **Balance remaining categories**: Adjust other segments (e.g., *Chart* to 25%, *Caption* to 10%) to fit the new total (100%).

By aligning the distribution with the task's goals and using data-driven methods, the new dataset will better support the intended application while maintaining robustness.

**Case: Grounding & Counting**

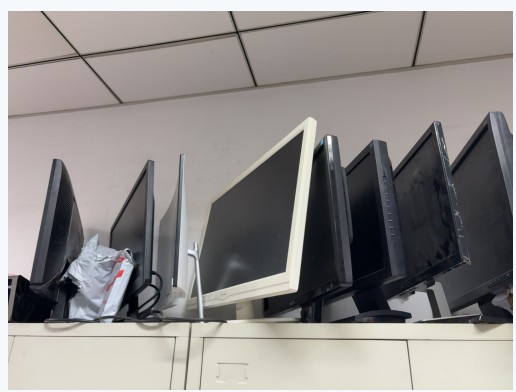

**User Question:**

How many monitors are there?

**Bee-8B Response:**

$< \text{think} >... < /\text{think} >$
To determine the number of monitors, we analyze the image by counting each distinct monitor from left to right:

1. The first monitor (leftmost, with a textured black front).

2. The second monitor (black).

3. The third monitor (silver).

4. The fourth monitor (white - bordered).

5. The fifth monitor (black, behind the white - bordered one).

6. The sixth monitor (black).

7. The seventh monitor (black).

8. The eighth monitor (rightmost, black).

Answer: 8

