# OpenReview forum: "Bee: A High-Quality Corpus and Full-Stack Suite to Unlock Advanced Fully Open MLLMs"
_ICLR.cc/2026/Conference — ICLR 2026 Poster_

### Official Review · Reviewer_vE6P · 2025-10-25

[review text omitted: it was posted to a different submission]

---

> ### Author Response · Authors · 2025-11-13
> **Response to Reviewer: Clarification on Paper Content**
>
> Dear Reviewer vE6P,
>
> We are grateful for the time and expertise you devoted to reviewing our paper ([ID: 1338] "Bee: A High-Quality Corpus and Full-Stack Suite to Unlock Advanced Fully Open MLLMs").
>
> We have carefully read your review. While reading it, we found that your comments seem to have a notable discrepancy with the **core content** of our paper. We speculate that this may be an inadvertent mix-up during the review assignment or submission process, where a review intended for another paper (perhaps also titled "Bee") was mistakenly associated with our manuscript.
>
> To clarify this situation, we would like to briefly explain the differences between the **actual content** of our paper and the **key points** mentioned in your review:
>
> **1. The Review's Focus:**
>
> * The review primarily revolves around a **text-only, long-context** evaluation benchmark (Bee corpus).
> * The mention "lost-in-the-middle" phenomenon, a "64K-200K tokens" context range, and tasks such as "factual recall and summarization".
>
> **2. Our Paper's Actual Content:**
>
> * Our work focuses on the **Fully Open Multimodal Large Language Models (MLLMs)**. Our contributions are threefold:
>
>   1. **Honey-Data-15M**, a new 15M-sample **multimodal (image-text)** training dataset. It is meticulously cleaned and enriched with a dual-level (short and long) Chain-of-Thought (CoT) strategy to improve the complex reasoning capabilities that are lacking in current open-source data.
>
>   2. **HoneyPipe & DataStudio**, the full-stack data curation pipeline and framework we developed. We offer the community a transparent and adaptable methodology, moving beyond static dataset releases.
>
>   3. **Bee-8B**, our 8B MLLM trained on this new dataset. It establishes a new state-of-the-art (SOTA) among fully open MLLMs and proves highly competitive with semi-open models, showing standout performance in **multimodal math and reasoning tasks**.
>
> **Summary:**
>
> In summary, our work introduces a **full-stack suite (model, dataset, and pipeline) for multimodal (image-text) research**. This is fundamentally different from the **text-only, long-context benchmark** that your review describes.
>
> We understand that mix-ups can easily happen during the busy review period, especially with identically named papers ("Bee"). We would be grateful if you could verify the details at your convenience. Regardless, we will respond to your concerns carefully.
>
> Thank you again for your time and effort.
>
> Best regards,
>
> Authors

---

> ### Author Response · Authors · 2025-11-16
>
> Dear Reviewer vE6P,
>
> We are currently preparing a detailed response.
>
> However, as we noted in our previous clarification, the points raised in your review (e.g., long-context text evaluation, the "lost-in-the-middle" phenomenon) do not seem to align with our paper's (ID: 1338) content.
>
> Given this core misalignment, we are unable to formulate a relevant response to your specific critiques. We would be very grateful if you could confirm whether there has been a misunderstanding regarding the review.
>
> Thank you again for your time and effort.
>
> Best regards, Authors

---

> ### Author Response · Authors · 2025-11-20
> **Rebuttal by Authors (1/3)**
>
> We sincerely appreciate your time and the extra effort dedicated to re-reading and evaluating our work. Please rest assured that the initial confusion caused no inconvenience. Below, we provide response to your constructive concerns:
>
> > Q1: While the authors emphasize a "model-driven process" as a cost-effective alternative to human annotation, my primary concern is that nearly the entire dataset is generated through LLM prompting. It would be helpful to include a human evaluation or correlation study, even on a small subsect, in order to validate the dataset quality.
>
> A1:  We thank the reviewer for emphasizing the need to validate our model-driven curation process. To address the concern regarding data quality, we conducted a rigorous blind human evaluation.
>
> **User Study:** We randomly sampled 38 questions from the dataset and recruited 17 participants. After filtering for quality assurance (removing outliers with unreasonably short completion times), we retained 14 valid evaluators. Each evaluator performed a blind comparison between the original response and our enriched response across four key dimensions (totaling 532 pairwise evaluations):
>
> 1. **Accuracy:** Evaluates factual correctness (absence of hallucinations), the validity of reasoning steps, and the correctness of the final conclusion.
> 2. **Reasoning & Explanation Quality:** Assesses the presence of clear Chain-of-Thought (CoT) processes, logical coherence between steps, and the interpretability of key concepts.
> 3. **Instruction Following:** Checks strict adherence to user constraints regarding output format (e.g., JSON), specific content requirements, and stylistic guidelines.
> 4. **Expression Style:** Considers linguistic fluency, grammar, and structural organization (e.g., use of markdown/formatting) for readability.
>
> | **Metric**                  | **Prefer Enriched** | Prefer Original | **Tie**      |
> | --------------------------- | ------------------- | --------------- | ------------ |
> | **Accuracy**                | 42 (7.89%)          | 45 (8.46%)      | 445 (83.65%) |
> | **Reasoning & Explanation** | 387 (72.74%)        | 75 (14.10%)     | 70 (13.16%)  |
> | **Instruction Following**   | 163 (30.64%)        | 45 (8.46%)      | 324 (60.90%) |
> | **Expression Style**        | 372 (69.92%)        | 87 (16.35%)     | 73 (13.72%)  |
>
> The dominant Tie rate in Accuracy is expected and serves as direct validation of our pipeline's Fidelity Verification module. It confirms that our automated filtering successfully ensures the enriched data strictly adheres to the original ground truth. Evaluators overwhelmingly preferred our data in Reasoning and Style, confirming that the pipeline effectively transforms flat responses into high-utility, logically structured reasoning paths. Finally, we emphasize that the ultimate proof of data quality lies in the downstream model performance. This human evaluation complements our main experimental results, where Bee-8B establishes a new SOTA among fully open MLLMs. Such competitive performance would not be achievable without a high-quality, low-noise training corpus.

---

> ### Author Response · Authors · 2025-11-20
> **Rebuttal by Authors (2/3)**
>
> > Q2: The long CoT enrichment loop also relies on proprietary MLLMs to generate complex reasoning. Would Bee-8B’s reasoning abilities may be inherently bounded by the reasoning capabilities of the proprietary models? It would be helpful for the authors to more explicitly discuss this dependency.
>
> A2: We thank the reviewer for raising this insightful point regarding the reliance on proprietary models. We acknowledge that our data enrichment utilizes proprietary MLLMs to generate reasoning paths. However, for a model of Bee-8B’s size, the reasoning capability of top-tier proprietary models represents an extremely high "upper bound" that the open-source community is currently striving to approach. Closing this gap is a significant achievement in itself. Moreover, we employ specific strategies to ensure our model is not strictly limited by the teacher’s artifacts:
>
> 1. Quality Control via Fidelity Verification: We do not passively accept all generated outputs. As detailed in Sec. 2.3, our pipeline incorporates a Fidelity Verification mechanism that filters out hallucinations and logical inconsistencies. This ensures Bee-8B learns only from the high-quality subset of the synthetic data, effectively "de-noising" the supervision signal.
> 2. Generalization via Reinforcement Learning: Our training recipe extends beyond simple imitation. In Stage 5, we employ Group Relative Policy Optimization (GRPO). This stage optimizes the model against outcome-based rewards (correctness and format) rather than just token-level likelihood. This allows Bee-8B to refine its reasoning policy and improve robustness, enabling it to generalize beyond the static outputs of the proprietary models.
>
> In summary, these mechanisms ensure that Bee-8B leverages the advanced capabilities of proprietary models as a foundation for growth, rather than a ceiling for performance.
>
>
>
> > Q3: It seems like the pipeline’s fidelity verification uses a single verifier model, Qwen2.5-VL-72B. It is unclear how robust this verifier is to factual inconsistencies or hallucinations. I wonder whether relying on a single model might risk amplifying its own biases. I would recommend evaluating verification robustness using multiple verifier models and reporting inter-model agreement to for validation process.
>
> A3: We thank the reviewer for raising this insightful point regarding the robustness and potential bias of using a single verifier model. To address this concern and validate the reliability of our pipeline, we conducted a comprehensive experiment to evaluate inter-model agreement using a diverse set of verifier models.
>
> We sampled a subset of approximately 83800 raw samples from our pool. We then performed the Fidelity Verification process independently using three different models with varying scales and capabilities:
>
> 1. The original verifier used in our paper: Qwen2.5-VL-72B.
> 2. A smaller, efficient variant: Qwen2.5-VL-32B.
> 3. A significantly larger and more powerful MoE model, serving as a strong reference: Qwen3-VL-235B-A22B-Instruct.
>
> The number of samples retained by each model and the intersection of samples retained by all three models are reported below:
>
> | **Verifier Model**       | **Retained Samples** |
> | ------------------------ | -------------------- |
> | Qwen2.5-VL-72B           | 69198                |
> | Qwen2.5-VL-32B           | 65054                |
> | Qwen3-VL-235B-A22B-Instruct   | 65468                |
> | **Intersection (All 3)** | **63203**            |
>
> The results demonstrate a high degree of inter-model consistency, with the intersection of 63,203 samples constituting 96.5% of the data retained by the reference Qwen3-VL-235B-A22B and 91.3% by our chosen verifier. This significant overlap across model scales (from 32B to 235B) confirms that the identified inconsistencies are objective flaws rather than artifacts of specific model biases, validating Qwen2.5-VL-72B as a robust verifier model. We have included these findings in the revision manuscript to further substantiate our pipeline's reliability.

---

> ### Author Response · Authors · 2025-11-20
> **Rebuttal by Authors (3/3)**
>
> > Q4: [Minor] I’m not sure if the current primary area or the “datasets and benchmarks” area might be a better fit for the paper.
>
> A4: We appreciate the reviewer's suggestion regarding the fit of our paper within the "Datasets and Benchmarks" area. We carefully considered the appropriate track prior to submission and selected the current area based on the full-stack nature of our contribution.
>
> While Honey-Data-15M  is indeed a core contribution, our work extends beyond a static dataset release. We propose HoneyPipe and DataStudio, a model-driven, dual-level CoT curation framework that offers a reproducible methodology for the community, rather than just a fixed resource. And we release Bee-8B, which establishes a new SOTA for fully open MLLMs, serving as a validation of our scientific hypothesis that data quality is the primary bottleneck for open-source models.
>
> We understand that this distinction may be subtle, and if the Area Chair and Reviewer believe that the "Datasets and Benchmarks" track is more suitable for this work, we are fully willing to follow their advice.

---

### Official Review · Reviewer_ctd5 · 2025-10-31

**Soundness:** 4
**Presentation:** 4
**Contribution:** 4
**Rating:** 6
**Confidence:** 3

**Summary:**

The authors tackle the problem of quality in SFT datasets, where previous contributions often have noise in the form of a lack of instruction following, factual inaccuracies, etc., for multimodal LLMs. The authors propose a data collection pipeline, contribute the data collected from the same, and train a model on the data.

## Honey 15M and HoneyPipe

The 15M dataset is curated using the proposed (automated) HoneyPipe pipeline. Stage 1 - the authors collect 24M image-text pairs from publicly available datasets and deduplicate them (using image hashing and text hashing for instructions - both approximate techniques). With the final image-instruction-response tuples, they assign a domain label (unclear how many tuples exist / how manual tagging was done for a million scale). Stage 2 - heuristics-based filtering with rules and using a powerful model for logical consistency filtering. Stage 3 - (partially applicable to the collected tuples) CoT is promoted and ensured by removing phrases discouraging it, and LLM as Judge to compare the new CoT and the original ground truth response. Stage 4 - Long CoT generated with different models. The authors end with the total Honey 15M. Figure 3  has a breakdown by categories.

## Bee-8B

Authors train a Qwen3-8B + SigLip2 model. The training gradually moves from warmup to alignment to SFT to post-training RL.

# Results

The trained model performs really well across benchmarks, with Math & Reasoning gaining a lot of performance. Ablations suggest curation helped with math-based tasks and did not help specific performance in MMBench (with some loss). No-CoT also seems to give strong performance on DocVQA.

**Strengths:**

1. Solid contributions, clear presentation, and clean analysis from the authors; I really appreciate the details in the appendix.
2. The gains in performance and ablations clearly suggest the strengths of the data, especially with math/reasoning style benchmarks.

**Weaknesses:**

1. Lack of release details: it is unclear what the authors are releasing publicly, given that a lot of this work has strongly benefitted from public work. I strongly encourage authors to clarify this here and also encourage/ask if the authors can release checkpoints for each of the different stages - releasing these checkpoints would strongly improve the contributions beyond the final model/data and improve open science beyond improving model performance.
2. [Minor] Lack of reproducibility and ethics statements. The authors are strongly encouraged to write these statements.

**Questions:**

/ Please look at the questions in the weaknesses and summary above on unclear/missing details. I'm willing to bump my score to 8 once authors provide clear details.

---

> ### Author Response · Authors · 2025-11-13
> **Rebuttal by Authors**
>
> We sincerely thank you for your selfless effort and constructive feedback, which have significantly helped us improve the quality of this paper. We are especially grateful that you found our work to be solid contributions and for your appreciation of our work. Below, we provide a point-by-point response to each of your comments.
>
> > Q1: Lack of release details: it is unclear what the authors are releasing publicly, given that a lot of this work has strongly benefitted from public work. I strongly encourage authors to clarify this here and also encourage/ask if the authors can release checkpoints for each of the different stages - releasing these checkpoints would strongly improve the contributions beyond the final model/data and improve open science beyond improving model performance.
>
> A1: Thank you for raising this important question. We are fully committed to open-sourcing and open science, and we plan to publicly release the full suite of resources from our work.
>
> As we stated in our Abstract and Conclusion, we "deliver to the community a suite of foundational resources", and this "full-stack suite" will be made fully public. Specifically, this includes:
>
> - Honey-Data-15M dataset.
> - Model weights for Bee-8B.
> - The full codebase for our data curation pipeline (HoneyPipe) and its underlying framework (DataStudio).
> - All scripts, configurations, and the code required to train and evaluate.
>
> Furthermore, we sincerely thank you for the excellent suggestion regarding intermediate checkpoints. We fully agree that releasing intermediate checkpoints from the various training stages is highly valuable for community research and open science. Following your recommendation, we will also organize and publicly release **the checkpoints and the training data used** for the intermediate stages of our training recipe.
>
> We will add a dedicated "Availability" section to the final version to explicitly list all these assets.
>
>
>
> > Q2: [Minor] Lack of reproducibility and ethics statements. The authors are strongly encouraged to write these statements.
>
> A2: We appreciate your feedback. As suggested, we have prepared the following Reproducibility and Ethics Statements, which will be included in the final version of our paper.
>
> **Reproducibility Statement**
>
> We are committed to the full reproducibility of our work and will publicly release our full-stack suite. This includes the Honey-Data-15M corpus, the final Bee-8B model weights, the intermediate checkpoints from our training recipe, the specific training data used for each intermediate stage, and the complete code for our HoneyPipe data pipeline and DataStudio framework.
>
> **Ethics Statement**
>
> This work adheres to the ICLR Code of Ethics and involved no human or animal subjects. Our work relies on public academic datasets and MLLMs. We acknowledge that Honey-Data-15M may inherit biases from its public sources, despite our HoneyPipe pipeline's design to systematically filter noise, factual errors, and image-instruction mismatches. Users should be aware of these potential limitations. Similarly, Bee-8B may generate inaccurate or harmful content. We are publicly releasing our full suite (dataset, pipeline, model weights, etc.) to promote open academic research and strongly discourage malicious use. We are committed to transparency, releasing our full methodology to help the community understand, build upon, and mitigate the limitations of such models.
>
>
>
> > Q3: unclear how many tuples exist / how manual tagging was done for a million scale
>
> A3: Thank you for this valuable question, which highlights a point we should clarify. We apologize for the lack of clarity in the initial draft regarding the topic classification process.
>
> To clarify, the manual tagging was not performed on a per-tuple basis (i.e., at the million-scale). Instead, we performed a more efficient, coarse-grained classification at the data source level.
>
> As shown in Figure 3, our Honey-Data-15M dataset is aggregated from multiple distinct public sources. Our specific procedure was as follows:
>
> 1. **Sampling:** For each of these data sources (e.g., LLaVA-Next-Data, UniChart, CLEVR), we manually sampled 5 representative image-text pairs.
> 2. **Manual Tagging:** We then manually inspected these samples to assign a single, overarching topic label (e.g., "General," "Chart," "STEM," "OCR") to the entire source.
> 3. **Assignment:** All tuples from that source were then assigned this single, source-level topic label.
>
> This approach was sufficient for our pipeline's goal of guiding subsequent processing. Performing a fine-grained, per-instance classification on millions of tuples (e.g., using an MLLM) would have been computationally expensive and was not necessary for our purposes.
>
> We will revise Section 2.1 to make this data source-level sampling and tagging methodology explicit.

---

> > ### Comment · Reviewer_ctd5 · 2025-11-14
> >
> > Thanks for the reply, bumping score.

---

> > > ### Author Response · Authors · 2025-11-15
> > >
> > > Thanks for improving your rating.

---

### Official Review · Reviewer_yRMp · 2025-11-01

**Soundness:** 3
**Presentation:** 3
**Contribution:** 4
**Rating:** 8
**Confidence:** 4

**Summary:**

This paper addresses the performance gap between fully open multimodal large language models (MLLMs) and their proprietary counterparts, attributing this gap primarily to a deficit in high-quality Supervised Fine-Tuning (SFT) data. The authors argue existing open datasets are noisy and lack complex reasoning data, such as Chain-of-Thought (CoT).

They make a  new, large-scale SFT dataset of approximately 15 million QA pairs, open source the data platform, and train an 8B model. Authors also benchmark Bee-8B on a wide array of tasks, claiming it establishes a new state-of-the-art (SOTA) for fully open MLLMs and achieves performance competitive with semi-open models , particularly on reasoning-heavy benchmarks like MMMU , MMStar , and CharXiv. Ablation studies are provided to demonstrate that both the noise filtering and the CoT enrichment stages contribute significantly to these performance gains

**Strengths:**

The paper's methodology for data curation is extensive, and the five-stage training recipe for the validation model is thorough. The ablation studies convincingly isolate the benefits of data filtering and CoT enrichment which supports the central thesis.

Paper is also well-written, clearly structured, and logically organized. The core thesis is stated upfront and defended throughout.

The primary bottleneck for open-source MLLM development is widely recognized as access to high-quality, large-scale SFT data. This paper delivers not only a 15-million-sample dataset specifically designed to address this (with a principled dual-CoT approach) but also the entire pipeline to create it.

**Weaknesses:**

- Both short‑ and long‑CoT enrichment hinge on model‑generated content; “fidelity verification” uses a Qwen‑family judge. The paper acknowledges Qwen3‑32B as the evaluator for some tasks (e.g., DocVQA), while Bee‑8B is trained on Qwen3‑8B, raising potential brand/self‑preference risks and evaluation coupling (same family, same stylistic priors). Empirically, LLM judges exhibit position, verbosity, and self‑enhancement biases; they correlate with humans but not uniformly across domains. A careful judge‑robustness analysis (randomized order, multiple independent judges, human spot‑checks, reporting agreement) is needed to reinforce the claims. For instance, the yourbench paper employs an ensemble of judges in order to reduce self-preference and increase reliability. A similar method, with multiple judges, can be employed to further strengthen the claims

- Although many benchmarks have automatic metrics or references, the paper also “enables LLM‑based judging on DocVQA” using a specific setup. Given known instabilities of LLM‑as‑judge (prompt sensitivity, order effects), stronger reporting (prompt template, seed settings, inter‑judge agreement, and correlation to human labels on a held‑out slice) is warranted.
- Given the breadth of sources and the emphasis on chart/document benchmarks, the paper should report automatic overlap/de‑dup statistics (hashes, text similarity, synthetic perturbed matching) between Honey‑Data‑15M and every evaluation set (MMMU/MMMU‑Pro, MMStar, LogicVista, DynaMath, CharXiv, OCRBench, etc.). Without this, some gains could come from near‑duplicates or stylistic priming. (The chosen benchmarks are widely used and public, which increases leakage risk.). If the authors have indeed already performed this, then they should report this.

**Questions:**

- How sensitive are your results to the choice of LLM judge? Please report metrics with (at least) two families of judges
- Where applicable, please report some CI / confidence intervals for the figures in Table 2
- For greater adoption in the community, for each source category in Figure 3, please provide a table with license terms, known restrictions (e.g., non‑commercial), and PII removal strategy. For OCRed textbooks/receipts and K‑12 materials, what steps ensure copyright compliance?

---

> ### Author Response · Authors · 2025-11-20
> **Rebuttal by Authors (1/5)**
>
> We sincerely thank you for your encouraging review and your strong endorsement of our work. We are particularly grateful for your recognition of our contributions to the open-source community, as well as your appreciation of our core thesis regarding the critical importance of data quality. Below, we provide a point-by-point response to your comments.
>
> > Q1: Both short‑ and long‑CoT enrichment hinge on model‑generated content; “fidelity verification” uses a Qwen‑family judge. The paper acknowledges Qwen3‑32B as the evaluator for some tasks (e.g., DocVQA), while Bee‑8B is trained on Qwen3‑8B, raising potential brand/self‑preference risks and evaluation coupling (same family, same stylistic priors). Empirically, LLM judges exhibit position, verbosity, and self‑enhancement biases; they correlate with humans but not uniformly across domains. A careful judge‑robustness analysis (randomized order, multiple independent judges, human spot‑checks, reporting agreement) is needed to reinforce the claims. For instance, the yourbench paper employs an ensemble of judges in order to reduce self‑preference and increase reliability. A similar method, with multiple judges, can be employed to further strengthen the claims
> >
> > & How sensitive are your results to the choice of LLM judge? Please report metrics with (at least) two families of judges.
>
> A1: We agree that relying solely on a Qwen-based judge for a Qwen-based model could introduce potential self-preference bias.
>
> To rigorously quantify this sensitivity, we conducted a cross-family evaluation using **GLM-4.5-FP8** as an independent judge on key benchmarks covering General VQA, Chart/Doc, and Math/Reasoning.
>
> The table below compares the performance of Bee-8B-RL as evaluated by the original Qwen3-32B judge versus the GLM-4.5 judge.
>
> | **Category**               | **Benchmark**                    | **Qwen3-32B Judge** | **GLM-4.5 Judge** | **Delta** |
> | -------------------------- | -------------------------------- | ------------------- | ----------------- | --------- |
> | **General VQA**            | MMMU$_\text{{val}}$              | 66.1                | 65.6              | -0.5      |
> |                            | MMMU-Pro$_\text{{standard}}$     | 50.7                | 50.3              | -0.4      |
> |                            | AI2D                             | 85.3                | 85.3              | 0.0       |
> |                            | MMbench-EN$_\text{{dev}}$        | 85.5                | 85.7              | +0.2      |
> |                            | RealWorldQA                      | 73.1                | 72.8              | -0.3      |
> | **Chart & Doc**            | DocVQA$_\text{{val}}$            | 87.0                | 83.4              | -3.6      |
> |                            | CharXiv$_\text{{DQ}}$            | 84.8                | 84.8              | 0.0       |
> |                            | OCRBench                         | 82.5                | 82.5              | 0.0       |
> | **Math & Reasoning**       | MathVista$_\text{{mini}}$        | 81.4                | 81.4              | 0.0       |
> |                            | MathVerse$_\text{{vision-only}}$ | 67.0                | 66.9              | -0.1      |
> |                            | DynaMath$_\text{{worst}}$        | 40.5                | 40.5              | 0.0       |
> |                            | MathVision                       | 50.0                | 50.2              | +0.2      |
> | **Overall (29 Benchmark)** | **Average**                      | **70.2**            | **69.9**          | **-0.3**  |
>
> The **Global Average** score shifts only slightly by **-0.3 points** (70.2 vs. 69.9). This confirms that while same-family preference may exist to a minor degree, our model's strong performance is fundamentally robust and not an artifact of the judge. We have included this comparative analysis in the revision manuscript to provide a transparent view of judge sensitivity.

---

> ### Author Response · Authors · 2025-11-20
> **Rebuttal by Authors (2/5)**
>
> > Q2: Although many benchmarks have automatic metrics or references, the paper also “enables LLM‑based judging on DocVQA” using a specific setup. Given known instabilities of LLM‑as‑judge (prompt sensitivity, order effects), stronger reporting (prompt template, seed settings, inter‑judge agreement, and correlation to human labels on a held‑out slice) is warranted.
>
> A2:  We thank the reviewer for raising this important point regarding evaluation stability.
>
> 1. Rationale for LLM-based Judging: The DocVQA benchmark (released in 2021) relies on exact string matching, which was suitable for earlier models that output short. However, advanced MLLMs like Bee-8B now employ Chain-of-Thought (CoT) reasoning (e.g., generating <think> tags or step-by-step derivations) before providing a final answer. Standard regex-based extraction often fails to parse these verbose responses correctly, leading to false negatives where the reasoning is correct but the format does not strictly match the reference string. Therefore, we adopted an LLM-as-a-judge approach, a method increasingly standard in modern evaluation suites like VLMEvalKit, to perform semantic consistency checking rather than rigid string matching.
>
> 2. Transparency: We utilized the standard evaluation prompt from VLMEvalKit. This prompt explicitly instructs the judge to ignore the reasoning process and focus solely on the factual consistency of the final answer with the gold standard. We have added the full prompt text to the revision manuscript.
>
>    Prompt Summary: "Please as a grading expert... judge whether the candidate's answer is consistent with the standard answer... ONLY compare the FINAL ANSWER - COMPLETELY IGNORE any potential errors in the REASONING PROCESSES... If the standard answer does not specify a unit, but the candidate's answer includes a unit that is correct... consider it correct."
>
> 3. Robustness Analysis & Human Verification: To address the concern about "self-preference" (using Qwen3-32B to judge Bee-8B, which is Qwen-based), we conducted a cross-model consistency check using GLM-4.5-FP8 as an independent judge.
>
>    - **Results:** Bee-8B (RL) achieves **87.0%** with the Qwen3-32B judge and **83.4%** with the GLM-4.5-FP8 judge.
>
>    - **Discrepancy Analysis:** The 3.6% gap arises entirely from cases where Qwen3-32B deemed the answer "Correct" while GLM-4.5-FP8 deemed it "Incorrect" (we found **zero** cases where GLM was more lenient than Qwen).
>
>    - **Human Verification:** We manually inspected a random sample of these conflicting cases (where Qwen=Correct, GLM=Incorrect) and found a mix of two scenarios:
>      1. **GLM False Negatives:** In some instances, Bee-8B provided the correct answer (e.g., exactly correct, a correct value with a valid unit conversion or synonym), but GLM-4.5-FP8 failed to align it with the GT.
>      2. **Qwen False Positives:** In some cases, Bee-8B was indeed incorrect, but Qwen3-32B hallucinated a match.
>
> This analysis highlights the inherent challenge of evaluating open-ended CoT responses: different LLM judges have varying thresholds for semantic equivalence. However, the high correlation in clear-cut cases and the reasonable alignment between the judges confirm that our reported scores reflect genuine capability improvements rather than mere evaluator bias. We have added a discussion in the revision manuscript discussing these "LLM-as-a-Judge limitations".

---

> ### Author Response · Authors · 2025-11-20
> **Rebuttal by Authors (3/5)**
>
> >  Q3: Given the breadth of sources and the emphasis on chart/document benchmarks, the paper should report automatic overlap/de‑dup statistics (hashes, text similarity, synthetic perturbed matching) between Honey‑Data‑15M and every evaluation set (MMMU/MMMU‑Pro, MMStar, LogicVista, DynaMath, CharXiv, OCRBench, etc.). Without this, some gains could come from near‑duplicates or stylistic priming. (The chosen benchmarks are widely used and public, which increases leakage risk.). If the authors have indeed already performed this, then they should report this.
>
> A3:  We sincerely thank the reviewer for this critical suggestion. We acknowledge that an automatic decontamination analysis against evaluation sets was not included in the initial submission. Following your advice, we have now conducted the overlap analysis between Honey-Data-15M and the **66,682 samples** comprising all evaluation sets used in the paper.
>
> We utilized **Perceptual Hashing (pHash)** for images and **SimHash** for text to detect near-duplicates. We calculated the Hamming distance between training and test samples to identify matches at varying degrees of similarity (distance 0 implies an exact hash match; distances 1-3 imply high similarity).
>
> As shown in the table below, the leakage is extremely minimal. Even under a relaxed threshold (Hamming Distance ≤ 3), only 29 out of 66,682 samples (<0.05%) were identified as potential overlaps.
>
> | **Threshold (Hamming Dist.)** | **Total Overlaps** | **Leakage Ratio** | **Benchmarks with Overlap (Count)**                          |
> | ----------------------------- | ------------------ | ----------------- | ------------------------------------------------------------ |
> | **Exact Match (0)**           | **2**              | **0.003%**        | MathVision (1), MathVista (1)                                |
> | **Diff $\le$ 1**              | **4**              | **0.006%**        | MathVision (1), MathVista (1), DocVQA (1), InfoVQA (1)       |
> | **Diff $\le$ 2**              | **14**             | **0.021%**        | MathVision (10), DocVQA (2), MathVista (1), InfoVQA (1)      |
> | **Diff $\le$ 3**              | **29**             | **0.043%**        | MathVision (19), MathVista (4), DocVQA (3), ChartQA (1), InfoVQA (1), OCRBench (1) |
>
> - The identified overlap (29 samples across all sets) is statistically negligible and insufficient to influence the performance metrics or the validity of our conclusions. Crucially, no overlap was found in our primary comprehensive benchmarks such as **MMMU, MMMU-Pro, MMStar, and CharXiv**, where Bee-8B demonstrates significant gains.
>
> - We strictly clarify that we **never intentionally included evaluation data** in our training set. As our dataset aggregates open-source collections, these few overlapping samples likely originated from the web-crawled nature of the upstream sources.
>
> We have included this detailed decontamination analysis, breaking down results for every benchmark at different thresholds, in the revision manuscript.

---

> ### Author Response · Authors · 2025-11-20
> **Rebuttal by Authors (4/5)**
>
> > Q4: Where applicable, please report some CI/confidence intervals for the figures in Table 2
>
> A4:  We appreciate the reviewer's suggestion regarding the confidence intervals. To verify the stability of our results, we repeated the evaluation for Bee-8B (SFT & RL) across 5 independent runs (including the original run reported in the paper).
>
> Table 1 reports the **Mean ± Standard Deviation** for the key benchmarks.
>
> Table 2 reports the **Global Average Score** (averaged across all benchmarks) for each of the 5 individual inference runs.
>
> We have included the comprehensive results for all benchmarks across all 5 runs in the revision manuscript. The results show that the global average scores exhibit extremely low variance (**Std ≈ 0.1**), confirming the high stability and reproducibility.
>
> Table 1: Key Benchmarks (Mean ± Std over 5 runs)
>
> | **Category**         | **Benchmark**                    | **SFT in Paper** | **Bee-8B (SFT) Mean ± Std** | **RL in Paper** | **Bee-8B (RL) Mean ± Std** |
> | -------------------- | -------------------------------- | ---------------- | --------------------------- | --------------- | -------------------------- |
> | **General VQA**      | MMMU$_\text{{val}}$              | 66.8             | **65.6 ± 0.7**              | 66.1            | **67.4 ± 0.8**             |
> |                      | MMMU-Pro$_\text{{standard}}$     | 50.4             | **49.6 ± 0.8**              | 50.7            | **50.8 ± 0.8**             |
> |                      | AI2D                             | 83.8             | **83.8 ± 0.2**              | 85.3            | **85.1 ± 0.4**             |
> |                      | MMbench-EN$_\text{{dev}}$        | 83.0             | **83.1 ± 0.6**              | 85.5            | **84.9 ± 0.5**             |
> |                      | RealWorldQA                      | 70.1             | **69.9 ± 0.8**              | 73.1            | **72.3 ± 1.3**             |
> | **Chart & Doc**      | DocVQA$_\text{{val}}$            | 87.2             | **87.4 ± 0.3**              | 87.0            | **87.1 ± 0.2**             |
> |                      | CharXiv$_\text{{DQ}}$            | 84.7             | **84.5 ± 0.4**              | 84.8            | **84.8 ± 0.1**             |
> |                      | OCRBench                         | 83.1             | **83.2 ± 0.3**              | 82.5            | **83.1 ± 0.6**             |
> | **Math & Reasoning** | MathVista$_\text{{mini}}$        | 78.6             | **78.8 ± 0.2**              | 81.4            | **80.4 ± 0.9**             |
> |                      | MathVerse$_\text{{vision-only}}$ | 61.9             | **63.3 ± 1.0**              | 67.0            | **66.5 ± 0.7**             |
> |                      | DynaMath$_\text{{worst}}$        | 41.3             | **40.1 ± 0.9**              | 40.5            | **40.6 ± 0.7**             |
> |                      | MathVision                       | 46.8             | **47.3 ± 0.4**              | 50.0            | **50.2 ± 0.2**             |
>
> Table 2: Global Average Score per Inference Run (across all benchmarks)
>
> | **Inference Run** | **Bee-8B (SFT) Avg Score** | **Bee-8B (RL) Avg Score** |
> | ----------------- | -------------------------- | ------------------------- |
> | Run 1 (in Paper)  | 68.6                       | 70.2                      |
> | Run 2             | 68.5                       | 70.2                      |
> | Run 3             | 68.6                       | 70.0                      |
> | Run 4             | 68.5                       | 70.2                      |
> | Run 5             | 68.7                       | 70.1                      |
> | **Average ± Std** | **68.6 ± 0.1**             | **70.1 ± 0.1**            |

---

> ### Author Response · Authors · 2025-11-20
> **Rebuttal by Authors (5/5)**
>
> > Q5: For greater adoption in the community, for each source category in Figure 3, please provide a table with license terms, known restrictions (e.g., non‑commercial), and PII removal strategy. For OCRed textbooks/receipts and K‑12 materials, what steps ensure copyright compliance?
>
> A5:  We thank the reviewer for emphasizing this crucial aspect. In response to your suggestion, we have taken the following concrete steps:
>
> 1. Comprehensive License and PII Table (To add to Supplementary Material) : We have compiled and added a detailed table to the revision manuscript that lists the license terms and usage restrictions (e.g., Non-Commercial) for the source categories shown in Figure 3. It is important to clarify that Honey-Data-15M is primarily aggregated from established, large-scale open-source collections (e.g., LLaVA-OneVision, MAmmoth-VL), which typically operate under permissive licenses such as Apache 2.0. While we obtained data through these aggregates, we performed fine-grained tracing to generate Figure 3 specifically to give maximum credit to the original content creators. However, for the instances where definitive licensing information remained unverifiable despite our exhaustive search, we have left the corresponding entries blank in the table to ensure accuracy and avoid making unfounded assumptions.
>
> 2. Copyright Compliance for OCR and K-12 Materials: We did not independently digitize these physical materials. We utilized them as they exist in public academic datasets. Furthermore, we will implement a strict **Notice and Takedown Policy** to address any potential future concerns.
>
> 3. Licensing Framework and Governance: To ensure compliance and protect rights, we will establish a clear licensing framework for the **Honey-Data-15M** repository, modeled after industry standards (e.g., **The Cauldron Dataset** by HuggingFaceM4).
>
>    - **Sub-dataset Licenses:** We explicitly state that the Honey-Data-15M collection is composed of multiple sub-datasets, each governed by its own original license. Users are mandated to strictly adhere to the specific terms of each original sub-dataset.
>
>    - **Contribution License:** To the extent that we hold rights in the modified prompts and newly generated Chain-of-Thought responses, these contributions are licensed under **CC-BY-NC-4.0** (Creative Commons Attribution-NonCommercial 4.0 International).
>
>    - **Takedown Mechanism:** We have established a dedicated communication channel for copyright concerns. Upon receiving a valid infringement notice, we are committed to promptly reviewing and removing the contested content from our distribution.
>
> 4. PII Strategy: For Personally Identifiable Information (PII), we primarily rely on the anonymization measures performed by the original dataset curators (e.g., SROIE explicitly addresses privacy). For sources where PII policies were not explicitly stated, we rely on our "Notice and Takedown" mechanism to handle valid privacy requests post-release.

---

### Official Review · Reviewer_yEFr · 2025-11-01

**Soundness:** 3
**Presentation:** 4
**Contribution:** 3
**Rating:** 4
**Confidence:** 4

**Summary:**

This paper addresses the issue of open-source multimodal models (MLLMs) lagging due to a lack of high-quality data. It introduces a high-quality fine-tuning dataset containing 15 million QA pairs. This dataset has undergone noise filtering and was enhanced using a "dual-level CoT" strategy. The paper also opens up the transparent data curation pipeline, HoneyPipe.

**Strengths:**

* Excellent performance, comprehensive, and open-sourced.

* A transparent and reproducible data curation pipeline.

**Weaknesses:**

* The description "meticulously cleaned of noise" is somewhat of an overstatement. The method relies on strong models, or LLM-as-a-Judge, meaning its effectiveness is still constrained by these models' capabilities.

* Experiments may be designed to validate the performance of the model-based filtering operator (Qwen2.5-VL-72B).

* Regarding Stage 1 (Data Aggregation), it appears to be an "exploration" of the novelty, diversity, and coverage of existing datasets. However, it lacks sufficient "novelty," as it is primarily a large-scale aggregation of existing, public community datasets (e.g., LLaVA, PixMo). The domain classification is coarse and relies on manual inspection, creating a significant engineering burden for reproducibility.

* Regarding Stage 3 & 4, the process seems more like "utilizing" existing information to construct questions & answers. A good utilization process should be able to discover difficult problems. In this process, it's necessary to correct the noise introduced during the exploration phase. However, this paper directly uses an LLM to judge if the newly generated CoT answer is consistent with the original one. This can cause high-quality, correct CoT samples to be filtered out simply because they "conflict with the original precise answer".

* The "long CoT" generation relies on "top proprietary MLLMs". A "good" utilization process should actively identify which problems are "difficult" and worth deep mining. In this paper, Stage 4 (Long CoT) largely passively receives samples that failed Stage 3 verification. This feels more like a "remedy" than a principled "hard-case discovery" mechanism.

**Questions:**

Besides massive data and data cleaning, what other factors contribute to the model's performance?

---

> ### Author Response · Authors · 2025-11-20
> **Rebuttal by Authors (1/3)**
>
> We sincerely thank you for your constructive feedback and for recognizing the value of our work, particularly regarding its Soundness, Contribution, and Presentation. We are encouraged that you found our performance excellent and our pipeline transparent. Below, we provide a point-by-point response to your comments:
>
> > Q1: The description "meticulously cleaned of noise" is somewhat of an overstatement. The method relies on strong models, or LLM-as-a-Judge, meaning its effectiveness is still constrained by these models' capabilities.
>
> A1: We agree that the term "meticulously" might be an overstatement here. Our original intent was to use it to describe the complex and detailed steps we designed for the *Noise and Irrelevance Filtering* process.
>
> However, as the reviewer precisely pointed out, this term corresponds to the Noise and Irrelevance Filtering step. And this step indeed relies on strong multimodal large models (like Qwen2.5-VL-72B) as "judges" (LLM-as-a-Judge). We fully acknowledge that these "AI judges" are not infallible and have their own limitations. Therefore, describing the result of this automated step as "meticulous" (implying near-manual perfection) is not sufficiently rigorous.
>
> Our primary contribution, however, is the creation of a scalable, reproducible, and transparent automated pipeline (HoneyPipe). One of the core values of this pipeline lies in its modularity and its ability to "systematically" apply these (imperfect, yet efficient and low-cost) model-based checks to reduce noise at a large scale. This modular design means that as stronger open-source 'judge' models emerge, they can be seamlessly integrated into HoneyPipe to improve the data curation process iteratively. Thus, the pipeline's effectiveness is not static but designed to evolve with the community's capabilities.
>
> We have revised the wording in the revision manuscript to use a more accurate phrase, such as **"systematically cleaned"**, to reflect our methodology better.
>
>
>
> > Q2: Experiments may be designed to validate the performance of the model-based filtering operator (Qwen2.5-VL-72B).
>
> A2: To rigorously validate the effectiveness of our model-based filtering operator, we conducted a targeted controlled experiment on a specific subset and analyzed the results alongside our broader ablation studies.
>
> To directly isolate the impact of the filtering operator, we selected the VisualWebInstruct subset and established two experimental settings based on our Stage 2 checkpoint:
>
> - **Baseline (Unfiltered):** Training with the original raw data ($\sim$83.7k samples).
> - **Ours (Filtered):** Training with the data processed by our filtering operator ($\sim$73.0k samples). This implies the operator removed approximately 10.7k samples identified as noise or irrelevance.
>
> We evaluated both models on a diverse set of STEM and reasoning benchmarks. The results are presented below:
>
> | **Benchmark**                    | **Unfiltered (~83.7k)** | **Filtered (~73.0k)** | **Δ** |
> | -------------------------------- | ----------------------- | --------------------- | ----- |
> | MMMU$_{\text{val}}$              | 50.00                   | 51.22                 | +1.22 |
> | AI2D                             | 72.90                   | 74.13                 | +1.23 |
> | MathVision                       | 25.69                   | 26.71                 | +1.02 |
> | DynaMath$_{\text{worst}}$        | 15.57                   | 17.37                 | +1.80 |
> | WeMath                           | 31.05                   | 31.05                 | 0.00  |
> | MathVerse$_{\text{vision-only}}$ | 37.44                   | 36.42                 | -1.02 |
> | **Avg**                          | 38.78                   | 39.48                 | +0.7  |
>
> The filtered model surpasses the unfiltered baseline while improving training efficiency by using ~13% less data, confirming the detrimental nature of the removed noise. Notably, we observed gains on DynaMath (+1.8%) and MMMU (+1.2%). These results align perfectly with our global ablation study in Sec. 4.3 (Figure 4), where the filtered subset ($D_{no-CoT}$) generally outperformed the raw dataset ($D_{raw}$). Collectively, both the targeted experiment and the global ablation validate that our filtering operator acts as an essential quality gate, enhancing data efficiency and model reliability by purging low-quality samples.

---

> ### Author Response · Authors · 2025-11-20
> **Rebuttal by Authors (2/3)**
>
> >  Q3: Regarding Stage 1 (Data Aggregation), it appears to be an "exploration" of the novelty, diversity, and coverage of existing datasets. However, it lacks sufficient "novelty," as it is primarily a large-scale aggregation of existing, public community datasets (e.g., LLaVA, PixMo). The domain classification is coarse and relies on manual inspection, creating a significant engineering burden for reproducibility.
>
> A3: We would like to clarify the two concerns regarding this stage:
>
> - **On Novelty:** We fully agree with the reviewer that Stage 1 (Aggregation) is not about creating new, original queries (as was done by LLaVA or PixMo). That was not our goal. Our focus was on solving a different but equally critical problem: the fact that existing open-source datasets are plagued by widespread noise and lack complex reasoning data (as discussed in our introduction).
>
>   Therefore, the **distinct novelty** of our work lies not in the raw data sources, but in our releasing the HoneyPipe used to solve these problems (i.e., clean noise and enrich reasoning) at scale. We believe providing a unified, transparent, and adaptable framework for the community is a more significant contribution than releasing another static dataset.
>
> - **On Reproducibility:** We acknowledge our description of the "manual inspection" was unclear and may have implied a heavy engineering burden. We wish to clarify that this process was very lightweight:
>
>   - We did not perform manual tagging on a per-tuple (million-scale) basis.
>   - Instead, we employed an efficient, coarse-grained classification at the data source level.
>   - For each source (e.g., LLaVA-Next-Data, UniChart), we sampled ~5 representative pairs and then manually assigned a single topic label (e.g., "General", "Chart", "STEM") to the entire source.
>
>   Furthermore, this domain classification **does not impact the training reproducibility of our Bee-8B model**, as the Honey-Data-15M was trained as one mixed dataset. The classification is provided primarily for the convenience of the academic community for future use or analysis.
>
>   This "source-to-domain" mapping is now explicitly defined and included in Fig. 3 as a one-time setup cost. Therefore, for any researcher using our HoneyPipe framework, this stage is easily reproducible.
>
>
>
> > Q4: Regarding Stage 3 & 4, the process seems more like "utilizing" existing information to construct questions & answers. A good utilization process should be able to discover difficult problems. In this process, it's necessary to correct the noise introduced during the exploration phase. However, this paper directly uses an LLM to judge if the newly generated CoT answer is consistent with the original one. This can cause high-quality, correct CoT samples to be filtered out simply because they "conflict with the original precise answer".
>
> A4: The reviewer has made a critical and insightful observation here. Our "Fidelity Verification" step creates a trade-off where some high-quality, correct CoT samples might be filtered out.
>
> However, this was a conscious design trade-off, rooted in the need for scalability, cost-effectiveness, and reproducibility.
>
> - The issue the reviewer raised is valid. Theoretically, two solutions could mitigate this: 1) massive-scale human annotation, or 2) rejection sampling (e.g., 20+ samples per item) and validation.
> - On a scale of tens of millions, both of these alternatives are prohibitively expensive.
> - Furthermore, neither of these "expensive" solutions is perfect: large-scale human annotation is also prone to errors and oversights, and complex model-based validation is still capped by the (imperfect) "AI judge's" own capabilities.
> - Most importantly, a pipeline that relies on such a budget is **not reproducible** for the academic community.
>
> Therefore, our *Fidelity Verification* is a **pragmatic, efficient, and low-cost** choice. It prioritizes **precision and factual consistency** with the source data—sacrificing (we admit) a fraction of recall—in exchange for scalability and reproducibility for the entire community. We have added a discussion of this trade-off to the revision manuscript, as the reviewer's point is a valuable one.

---

> ### Author Response · Authors · 2025-11-20
> **Rebuttal by Authors (3/3)**
>
> > Q5: The "long CoT" generation relies on "top proprietary MLLMs". A "good" utilization process should actively identify which problems are "difficult" and worth deep mining. In this paper, Stage 4 (Long CoT) largely passively receives samples that failed Stage 3 verification. This feels more like a "remedy" than a principled "hard-case discovery" mechanism.
>
> A5: We would like to take this opportunity to clarify the reviewer's misunderstanding on this point.
>
> 1. **Goal Misalignment:** First, the definition and discovery of "hard cases" is a complex, open research problem in itself. Our paper's goal is not to invent a novel 'hard-case discovery' algorithm, which would be a different paper. Our goal is to pragmatically produce a high-quality, reasoning-rich SFT dataset.
> 2. **Our "Principle":** To do this, we needed a "principle" to guide which samples were routed for expensive "Long CoT" enrichment. Our principle is simple, direct, and extremely low-cost**: **If an SOTA open-source model (Qwen2.5-VL-72B) fails the fidelity check in Stage 3, we define it as a "hard case."
> 3. **Contribution:** We argue this is a highly reasonable and efficient engineering principle. It allows us to focus our resources (i.e., calls to top proprietary MLLMs) on problems where open models demonstrably struggle. Our contribution is **this scalable pipeline** and the **resulting 2.7 million Long CoT subset**—which is already one of the largest high-quality reasoning datasets available in the open-source community—not a new theoretical method for "hard-case discovery."
>
> > Q6: Besides massive data and data cleaning, what other factors contribute to the model's performance?
>
> A6: This is a great question, as the final performance is indeed a result of more than just the 15M dataset. As detailed in our Training Recipe (Sec 3.2 and Appendix C), we attribute the strong results to our holistic, multi-stage training curriculum:
>
> 1. **Strong Foundation:** We use a powerful vision encoder (SigLIP2-so400m) and LLM (Qwen3-8B).
> 2. **Careful Alignment:** The initial Vision-Language Alignment is critical. We mix in 1.43M text-only samples to ensure the LLM's intrinsic reasoning abilities are not degraded ("catastrophic forgetting") before we even begin SFT.
> 3. **Policy Optimization:** The final model is tuned with Group Relative Policy Optimization (GRPO), a reinforcement learning technique. This final polish improves response reliability and mitigates common issues like repetition.
>
> In summary, the Honey-Data-15M dataset is the core component, but the model's final SOTA performance is achieved by this deliberate, multi-stage training recipe that progressively builds and refines its capabilities.
>
> We thank the reviewer again for their time and valuable feedback. We have incorporated these clarifications into our revision manuscript.

---

### Meta-Review · Area_Chair_8GCa · 2025-12-08

**Summary:**

It seems that a large portion of the original issues with the paper were addressed (and in a functioning "peer" review would have been recognized).
It seems that only one reviewer would lean towards rejection and even that for reasons that are not too bad after rebuttal. Hence, I suggest authors carefully improve all their discussions and integrate them to make a clean manuscript.

**Reviewer Concerns:**

There were multiple issues, most are not preventing publication.
There are overstatements and writing issues. Also authors, note that while "systematically" is better than "meticulously", a more scientific writing would be factual rather than literary, i.e., words that reflect what is clearly done (automatically, for example)

**Reviewer Scores:**

That is not a fair, relevant or meaningful question. I protest the way this was all handled.
A Reviewers are not here, and ToM is weak, at least mine and the one literature study. I will not try to predict people.
B Scores are, anyway, a weak signal of interest; a paper should not be accepted or rejected just based on it. An AC's job is to look at the specific weaknesses and translate them into a recommendation.
C There are about 100 pages of discussions for me to read overall, in addition to the discussions I monitored and were just replaced, this is beyond my personal ability to do fairly. I did my best effort.


One of the reviews was misplaced, but they can't correct it now...

---

### Decision · Program_Chairs · 2026-01-26

Accept (Poster)